# Li$_{1.5}$La$_{1.5}$$M$O$_6$ ($M=$ W$^{6+}$, Te$^{6+}$) as a new series of lithium-rich double perovskites for all-solid-state lithium-ion batteries

Marco Amores [1], Hany El-Shinawi[1,2], Innes McClelland [1], Stephen R. Yeandel [3], Peter J. Baker [4], Ronald I. Smith [4], Helen Y. Playford [4], Pooja Goddard [3], Serena A. Corr [1,5✉] & Edmund J. Cussen [1,5✉]

Solid-state batteries are a proposed route to safely achieving high energy densities, yet this architecture faces challenges arising from interfacial issues between the electrode and solid electrolyte. Here we develop a novel family of double perovskites, Li$_{1.5}$La$_{1.5}$$M$O$_6$ ($M=$ W$^{6+}$, Te$^{6+}$), where an uncommon lithium-ion distribution enables macroscopic ion diffusion and tailored design of the composition allows us to switch functionality to either a negative electrode or a solid electrolyte. Introduction of tungsten allows reversible lithium-ion intercalation below 1 V, enabling application as an anode (initial specific capacity >200 mAh g$^{-1}$ with remarkably low volume change of ~0.2%). By contrast, substitution of tungsten with tellurium induces redox stability, directing the functionality of the perovskite towards a solid-state electrolyte with electrochemical stability up to 5 V and a low activation energy barrier (<0.2 eV) for microscopic lithium-ion diffusion. Characterisation across multiple length- and time-scales allows interrogation of the structure-property relationships in these materials and preliminary examination of a solid-state cell employing both compositions suggests lattice-matching avenues show promise for all-solid-state batteries.

[1] Department of Chemical and Biological Engineering, University of Sheffield, Sheffield S1 3JD, UK. [2] The Faraday Institution, Harwell Campus, Didcot OX1 0RA, UK. [3] Department of Chemistry, Loughborough University, Epinal Way, Loughborough LE11 3TU, UK. [4] ISIS Pulsed Neutron and Muon Source, STFC Rutherford Appleton Laboratory, Harwell Science and Innovation Campus, Didcot, Oxfordshire OX11 0QX, UK. [5] Department of Materials Science and Engineering, University of Sheffield, Sheffield S1 3JD, UK. ✉email: s.corr@sheffield.ac.uk; e.j.cussen@sheffield.ac.uk

Accessing high performance all-solid-state Li-ion batteries remains an outstanding grand challenge in the battery research community. The desire to move towards an all solid battery configuration is driven by safety concerns, enabling the use of metallic lithium anodes and the replacement of flammable liquid electrolytes used ubiquitously in existing Li-ion batteries[1,2]. These liquid electrolytes are readily flammable and chemically unstable at high voltages and temperatures, which compromises the safety of the battery[3–5]. All-solid-state Li batteries are a promising alternative, which could overcome not only these pressing safety concerns but would also see an increase in the achievable energy density in these Li batteries by extending the potential window to permit the safe use of high-voltage cathodes and metallic lithium anodes[6]. However, a lack of reliable solid-state electrolytes is currently hampering the development of all-solid-state batteries and complexities of delivering high Li-ion diffusion at the electrode–electrolyte interface mean that there is a pressing need for new families of functional solid-state materials to fulfil this role[7–9].

To address this challenge, we have turned to the perovskite family of compounds, where the prototype cubic structure possesses the formula unit $ABO_3$, with $A$-site metal cations 12-coordinated to oxygen and $B$-site cations octahedrally coordinated by 6 oxygen atoms. Oxide perovskite materials are versatile materials with an impressive range of applications due to their exotic physical properties, including ferroelectric, dielectric, pyroelectric, and piezoelectric behaviours[10]. This versatility owes much to the robust framework, which permits multiple combinations of different cations and anions to be present in the structure, making this an ideal candidate for battery applications as it can accommodate a wide range of cation sizes and oxidation states. Lower-symmetry structures can also be obtained by variations in the relative sizes of the $A$ and $B$ cations[10,11]. Our intention is to derive a family of compounds where judicious choice of the $B$-site cation can lead to changes in conductivity, allowing for the design of active electrode and solid-electrolyte materials of the same structure type. Our ultimate ambition is to realise a family of lithium-containing materials where chemical compatibility and ion mobility at the electrode–electrolyte boundary are maximised by shared crystal structure and minimal changes in compositional variation across this interface.

Lithium is most commonly accommodated in the perovskite structure in the oxide octahedra. However, there is an extensively studied class of compounds where lithium is introduced into the larger site to give stoichiometries such as $(La,Li)TiO_3$. These materials continue to attract great interest owing to reports of fast ion conduction, e.g., in $Li_{0.34}La_{0.5}TiO_{2.94}$[12,13]. This arises from displacement of the lithium from the centre of the perovskite cube to occupy a four-coordinated, distorted square-planar site. An extensive compositional search reveals that both the structural arrangement and the ionic conductivity are highly anomalous and the fast conduction of $Li_{0.34}La_{0.5}TiO_{2.94}$ has not been reproduced in analogues based on replacement of $La^{3+}$ with other rare earth cations or alkaline earth elements or by substitution of $Ti^{4+}$ with $Zr^{4+}$, $Nb^{5+}$, $Ta^{5+}$ or other appropriate cations[14–17].

Despite the multiple applications of perovskite materials, their use in Li-ion batteries is limited to only a few reports, namely, lithium lanthanum titanate as a fast lithium conductor and lithium lanthanum niobate as an insertion electrode[13,18]. Introduction of a second cation on the $B$ site can introduce additional complexity from chemical ordering of these cations over different crystallographic sites in so-called double perovskite structures as shown in Fig. 1[17]. We have serendipitously discovered a lithium tungstate double perovskite, which has also been independently identified by another group[19].

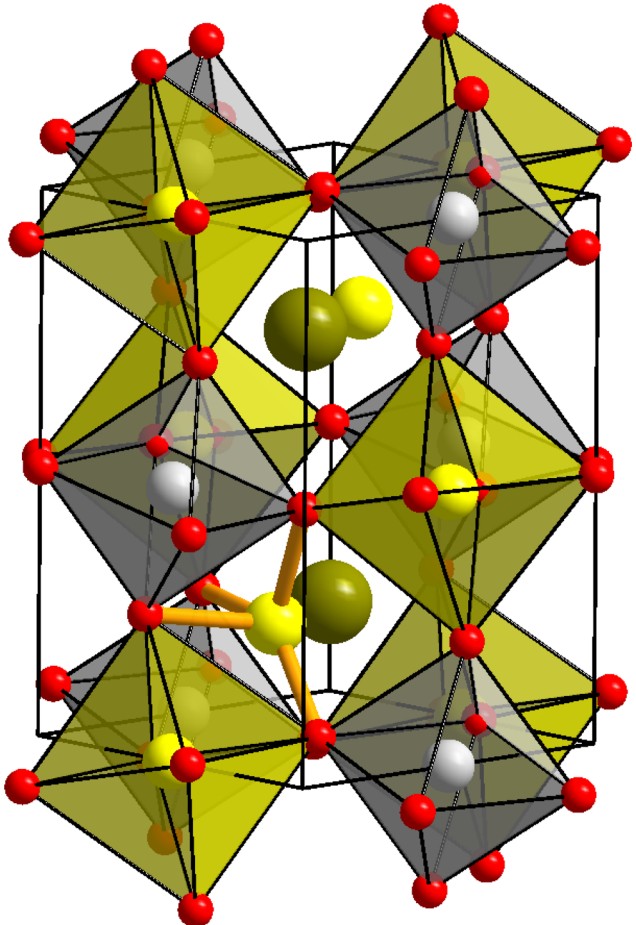

**Fig. 1 Crystal structure of Li$_{1.5}$La$_{1.5}$MO$_6$ double perovskites.** Cation ordering across octahedral sites occurs as shown here in the monoclinically distorted double perovskite structure of $Li_{1.5}La_{1.5}WO_6$. Grey spheres in the octahedra represent $W^{6+}$ ions; dark green spheres represent $La^{3+}$ ions at the centre of the large $A$-site interstice. Lithium cations are shown as yellow spheres occupying half of the oxide octahedra in a rock salt ordering pattern. The remaining lithium cations occupy four-coordinated sites that are displaced from the centre of the large $A$-site interstice. The coordination environment for one $LiO_4$ unit is indicated by orange bonds.

Here we demonstrate for the first time that the combination of lithium mobility along with a redox active metal in a high oxidation state allows for the electrochemical (de)intercalation of lithium to provide a new class of electrode materials for Li-ion batteries. Since the perovskite structure is famously amenable to chemical and structural adjustment, we propose that this is the first in a new class of perovskite lithium electrode materials. To further demonstrate this chemical adjustment and how it pertains to property tuning, we also report a tellurium analogue, which displays promising solid-electrolyte capabilities. As well as their comprehensive physical and chemical characterisation, we also demonstrate the first proof-of-concept measurements of their combined performance in a hybrid solid-state Li-ion battery.

## Results

**Synthesis, structural characterisation and chemical composition.** The syntheses of novel lithium-rich double perovskites $Li_{1.5}La_{1.5}MO_6$ (where $M = W^{6+}$, $Te^{6+}$) used a microwave-assisted solid-state approach. Such synthetic methods have been developed extensively in our research group in recent years and afford faster, lower temperature routes to high quality solid-state

materials[20–23]. Optimisation of the reagent stoichiometries revealed that ratios of Li:La:$M$ of 1.5:1.5:1 were required for phase pure compounds ($M$ = $W^{6+}$, $Te^{6+}$). Ratios lying outside of these values consistently delivered impure products as shown by X-ray diffraction (XRD) data in Supplementary Fig. 1. Diffraction data from three target compositions $Li_{1.5}La_{1.5}WO_6$, $Li_{1.5}La_{1.5}TeO_6$ and $Li_{1.5}La_{1.5}W_{0.5}Te_{0.5}O_6$ could readily be indexed using a monoclinically distorted variant of the perovskite structure. This distortion commonly arises from a combination of cation ordering over the octahedral sites and a tilting of the oxide octahedral to reduce the size of the large $A$-site interstice to better match the bonding requirements of $A$ cations that are smaller than optimal. $Li^+$ cations are well matched in size to the $B$-site octahedral sites as shown in the double perovskites $La_2LiMO_6$[17,24]. The stoichiometries $Li_{1.5}La_{1.5}MO_6$ can be re-written as $(La_{1.5}Li_{0.5})(LiM)O_6$ to emphasise the relation to the classic $ABO_3$ perovskite formula. This description implies that $Li^+$ cations occupy both $A$ and $B$ sites of the structure, and this hypothesis was tested experimentally with X-ray and neutron powder diffraction (NPD). Using these two radiations in a simultaneous refinement allows the X-ray data to locate dominant X-ray scatterers, i.e. $La^{3+}$, $W^{6+}$ and $Te^{6+}$, and the neutron diffraction locates the lighter $Li^+$ and $O^{2-}$. Crucially, the complementary weightings of atomic number (X-rays) and scattering length (neutrons) provides contrast that allows us to search for vacancies in these structures.

Rietveld refinement for $La_{1.5}Li_{1.5}WO_6$ showed that the $A$ site is $ca.$ $^3/_4$ occupied by $La^{3+}$. The monoclinic distortion reduces the La coordination number to 8, from 12 in the undistorted structure. The $A$ site is also partially occupied in a disordered manner by $Li^+$ ions that are displaced towards the base of a trigonal pyramid (Fig. 1). This lowers the coordination number from eight for $La^{3+}$ to four for $Li^+$, and the shortening of the Li⋯O distances to values more typical of the smaller $Li^+$ cation. Trial refinements indicated that all oxide positions were within one esd, i.e. 1%, of being fully occupied so these were subsequently fixed at this value. All other atomic fraction parameters and thermal parameters were allowed to refine freely leading to the stoichiometry $La_{1.43(12)}Li_{1.39(4)}W_{0.97(1)}O_6$ (Supplementary Table 1), within three esd of the target stoichiometry. Elemental analyses by energy-dispersive analysis of X-rays and mass spectrometry were both in agreement with this (Supplementary Tables 2 and 3). The composition of this material will be referred to as $Li_{1.5}La_{1.5}WO_6$. The structure of $Li_{1.5}La_{1.5}WO_6$ was also assessed by X-ray absorption spectroscopy (XAS) using the W $L_{III}$-edge. This showed that the average W–O bond length and the W–La distance calculated from the EXAFS were in agreement with the diffraction analysis (Supplementary Fig. 2), indicating that the crystallographic structure is representative of the local configuration.

Replacement of tungsten with tellurium resulted in a material that gave X-ray and neutron diffraction profiles that could be indexed using the same $P2_1/n$ distortion. Simultaneous refinement of the structure against the X-ray and neutron diffraction profiles showed agreement with the structural model, but with a significant difference between the observed and calculated neutron diffraction profiles indicated by the relatively high value of $\chi^2 = 8.4$. This structural refinement was subjected to extensive testing for disorder in both Li/La and Li/Te arrangements, and insertion of $Li^+$ onto additional interstitial sites; all of which indicated that the cation arrangement in $Li_{1.5}La_{1.5}TeO_6$ was the same as the tungstate analogue, and the cation occupancies were subsequently fixed at the targeted composition. Difference Fourier searches indicated that the intensity mismatch was arising from shortcomings in the description of the oxide ions; when these displacements were modelled anisotropically, nonphysical values resulted. The fit was greatly improved by allowing

each of the crystallographic oxide ion positions to be split over two sites, with the fractional occupancy of each site initially refining freely. This resulted in six crystallographically independent oxide ion positions, with the occupancy of the sites showing that each position had been split in an approximate 1:2 ratio (Supplementary Fig. 3). For the final refinement, the oxide ions were paired such that a single site occupancy was used to model this disorder giving the structural parameters in Supplementary Table 4. This had negligible impact on the quality of the fit compared to the unconstrained disorder in the oxide occupancies, as shown in Fig. 2 with $\chi^2 = 5.35$. To investigate the possibility of a solid solution between both compositions, the mixed $Li_{1.5}La_{1.5}W_{0.5}Te_{0.5}O_6$ double perovskite was also synthesised. XRD analysis indicates the same $P2_1/n$ space group, with a unit cell volume of 246.704(9) $Å^3$ that is intermediate between $Li_{1.5}La_{1.5}WO_6$ [245.016(3) $Å^3$] and $Li_{1.5}La_{1.5}TeO_6$ [247.24(1) $Å^3$] (Supplementary Fig. 4). Raman spectra revealed the expected vibrational modes[25] were present in all three compounds, with both Te–O and W–O vibrations present in the $Li_{1.5}La_{1.5}W_{0.5}Te_{0.5}O_6$ material (Supplementary Fig. 5). The compositions of the three targets were further confirmed by energy-dispersive X-ray (EDX) and ICP elemental analyses (Supplementary Tables 2 and 3). We recently reported the Na-analogue $Na_{1.5}La_{1.5}TeO_6$ indicating of the wide versatility of these novel families of alkali metal-rich double perovskites[23]. The microstructures of $Li_{1.5}La_{1.5}WO_6$ and $Li_{1.5}La_{1.5}TeO_6$ feature quasi-spherical particles fused together by tubular joints as shown in Fig. 2. A larger particle size was noted for the tungsten compound (5–10 μm) compared with the tellurium analogue (1–5 μm).

**Electrochemical and ion transport properties.** The presence of $W^{6+}$ ions and the robust perovskite framework suggest this material may function as an insertion anode material, due to the accessible tungsten redox couples. To evaluate this possibility, cyclic voltammetry (CV) analyses were performed in the voltage range of 0.01–2.8 V against Li metal as the counter electrode, at a scan rate of 0.1 mV s$^{-1}$. In the first cycle, a broad peak in the reduction regime is observed due to the formation of a solid-electrolyte interphase (SEI), a consequence of a partial decomposition at low voltages of the carbonate liquid electrolyte and the high surface area of the carbon black used[26,27]. $Li_{1.5}La_{1.5}WO_6$ displays two strong redox couples (Fig. 3a). The first voltage redox couple occurs at ~0.7 V in the reduction regime and at ~1 V in the oxidation regime. The second redox process takes place at lower potentials; in the reduction regime the peak below 0.2 V is partially obscured by the broad peak near 0 V arising from lithium insertion into the carbon black additive[27], and near 0.3 V for the oxidation process. Interestingly, in the case of the isostructural $Li_{1.5}La_{1.5}TeO_6$ analogue, there are no reduction or oxidation peaks over the same potential range (Fig. 3b), apart from those arising from the carbon black additive (Supplementary Fig. 6). The intermediate phase $Li_{1.5}La_{1.5}W_{0.5}Te_{0.5}O_6$ (Supplementary Fig. 7) displays similar peak positions to the W parent compound, but with reduced current presumably from lower concentration of redox active W cations.

The absence of redox activity in $Li_{1.5}La_{1.5}TeO_6$ indicated that this compound could be redox stable in the potential window of a battery and suggested applications as a solid Li-ion electrolyte. Macroscopic ionic conduction was analysed for both materials by means of electrochemical impedance spectroscopy (Fig. 4), revealing activation energies for ionic conduction of 0.59(3) and 0.68(2) eV (Fig. 5) and ionic conductivities of $1.1 \times 10^{-5}$ and $5.8 \times 10^{-5}$ S cm$^{-1}$ at 124 °C for the W and Te materials, respectively (Supplementary Table 5). These conductivities are intermediate between the fast conducting garnets[28–31] based on

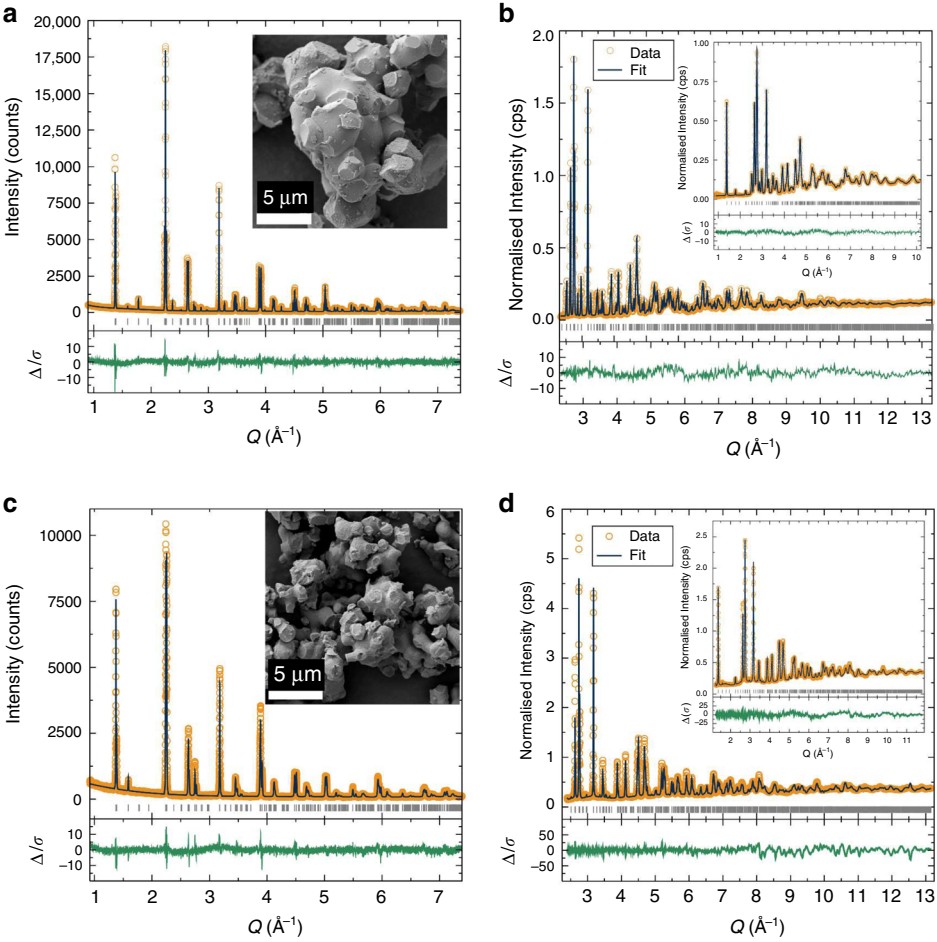

**Fig. 2 Crystal structure Rietveld refinements and particle morphology. a** X-ray and **b** neutron powder diffraction data for the $Li_{1.5}La_{1.5}WO_6$ material shown in the inset to (**a**). **b** Neutron diffraction data are shown over the combined Q-range 1–13 Å$^{-1}$ using two detector banks. **c, d** Corresponding data from $Li_{1.5}La_{1.5}TeO_6$. Fits were in good agreement to the monoclinic space group $P2_1/n$, with a partially disordered arrangement of oxide anions in $Li_{1.5}La_{1.5}TeO_6$. The SEM images reveal smaller particle sizes for the Te compound compared to the W analogue.

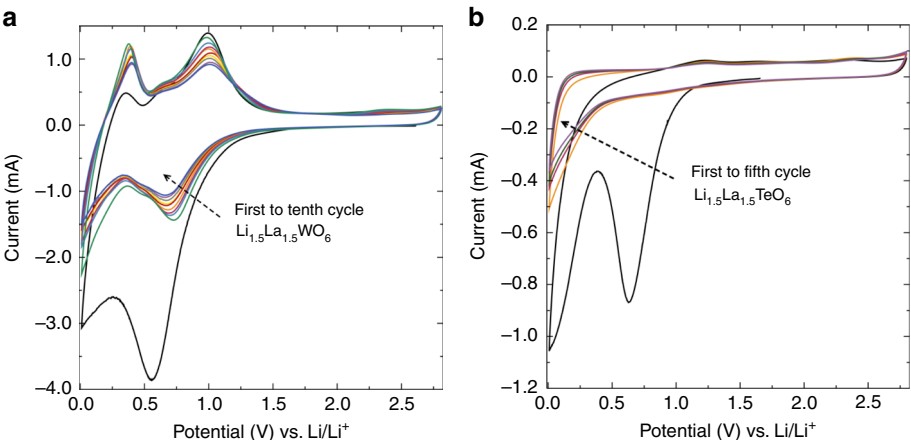

**Fig. 3 Redox response dependence of $M^{6+}$ cation. a** CV data for $Li_{1.5}La_{1.5}WO_6$ and **b** $Li_{1.5}La_{1.5}TeO_6$ materials mixed with 5% carbon black and 5% PTFE binder between 0.01 and 2.8 V vs Li. The scan rate was fixed at 0.1 mV s$^{-1}$ for both measurements.

$Li_7La_3Zr_2O_{12}$ and the lower conductivity of conventional garnets[32,33], such as the $Li_3Ln_3Te_2O_{12}$, that have the same cation ratio as these perovskites. It should be noted that these conductivities derive from sintered cold pressed pellets and include contributions from both intra- and inter-grain conduction. It is widely observed across garnets and other families of

ionically conducting oxides that compositional adjustments may dramatically increase intra-grain conduction, whilst inter-grain resistance may be greatly reduced by systematic adjustments to processing conditions, e.g., hot pressing or spark-plasma sintering (SPS)[34,35]. The Li-rich family of materials presented here are prime candidates for similar developments, with the added

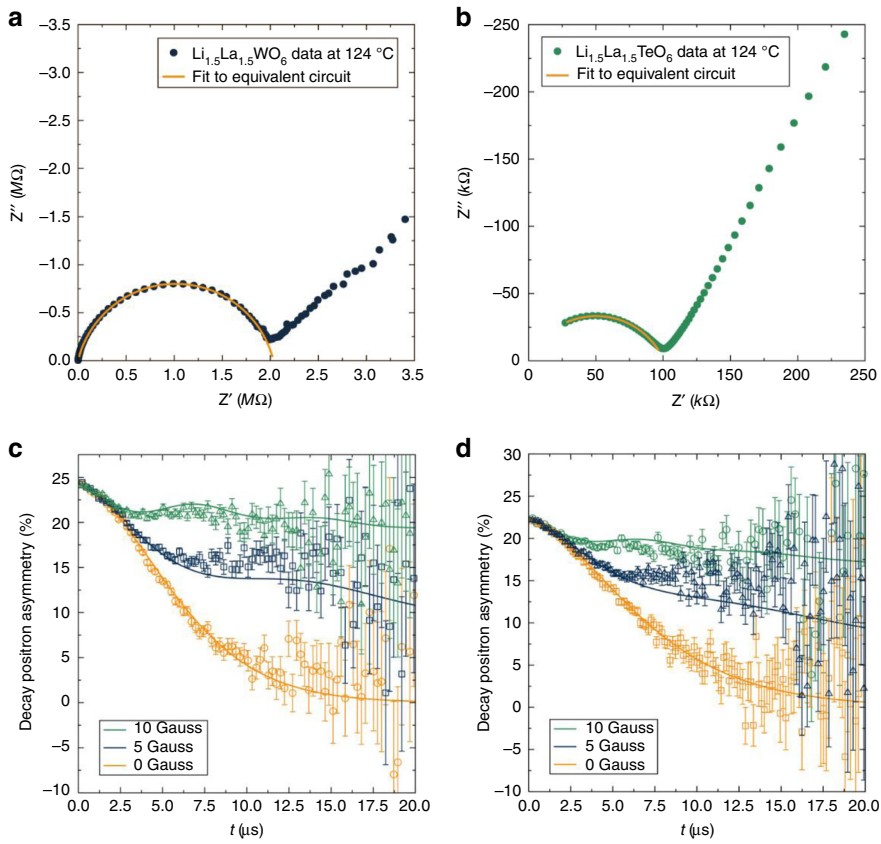

**Fig. 4 Macroscopic and local transport properties. a** Electrical impedance for $Li_{1.5}La_{1.5}WO_6$ and **b** $Li_{1.5}La_{1.5}TeO_6$ at 124 °C fitted to an equivalent electrical circuit of a resistor in parallel with a constant phase element. **c** μ⁺SR experiments show the temporal evolution of the decay positron asymmetry for $Li_{1.5}La_{1.5}WO_6$ and **d** $Li_{1.5}La_{1.5}TeO_6$ at room temperature, with the fits to the Keren function[40] indicated by solid lines. The error bars are indicated by raising lines on the same colour of the data set and with triangular, square or circular shapes for 10, 5 and 0 Gauss, respectively.

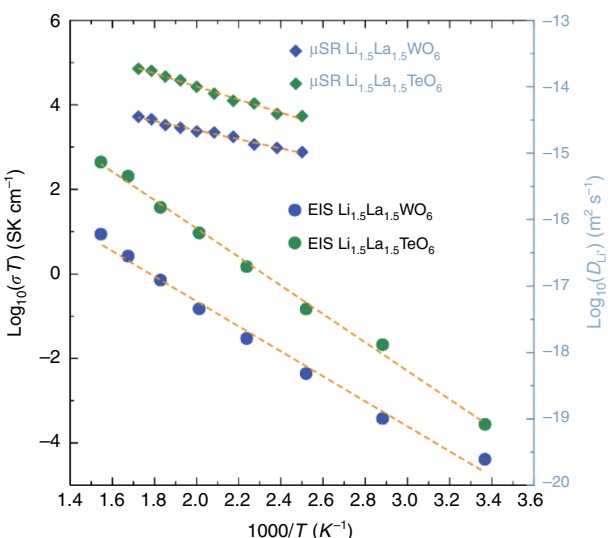

**Fig. 5 Arrhenius behaviour of conduction properties at macro and micro-scopic level.** Arrhenius plots of the macroscopic and microscopic ionic transport properties measured by electrical impedance spectroscopy (conductivity, left axis) and μ⁺SR (diffusivity, right axis) for $Li_{1.5}La_{1.5}WO_6$ and $Li_{1.5}La_{1.5}TeO_6$.

benefit that these afford both electrode and solid electrolyte with the same crystal structure and therefore present an opportunity for greater interfacial compatibility, a key consideration currently hampering the development of solid-state batteries[7,36].

The total conductivity measured by impedance spectroscopy can be complemented by muon spin relaxation (μ⁺SR) in assessing diffusion behaviour over a local scale of a few nanometres[37]. Spin-polarised muons are implanted into the material and the spin direction is perturbed by the passage of diffusing species, such as $Li^+$, that possess a nuclear moment. Analysis of the muon depolarisation as a function of temperature, combined with knowledge of the crystallographic distribution of $Li^+$, allows extraction of values of room temperature $Li^+$ diffusion coefficients of $6.6 \times 10^{-12}$ cm² s⁻¹ and $1.8 \times 10^{-11}$ cm² s⁻¹ for the W and Te materials, respectively, as shown in Figs. 4 and Supplementary Fig. 8. These values are similar to those obtained for other fast ion conductors studied using μ⁺SR, including the doped garnet materials based on $Li_7La_3Zr_2O_{12}$ ($4.62 \times 10^{-11}$ cm² s⁻¹), the anode material $Li_4Ti_5O_{12}$ ($3.2 \times 10^{-11}$ cm² s⁻¹) and the high energy NMC cathode ($3.5 \times 10^{-12}$ cm² s⁻¹)[21,38,39]. Arrhenius analyses, shown in Fig. 5, deliver activation energies of 0.136(5) for the W compound and 0.196(8) eV for Te analogue. These values are also similar to those obtained for $Na^+$ diffusion in our recently reported analogous Na-rich double perovskite, $Na_{1.5}La_{1.5}TeO_6$, of $4.2 \times 10^{-12}$ cm² s⁻¹ and 0.163(9) eV[23], indicating the versatility of the double perovskite framework for other battery chemistries beyond lithium. Our values from μ⁺SR experiments are in agreement with DFT calculations and NMR measurements previously reported on the tungstate[19] and are also in line with those of the related $(Li,La)TiO_3$ perovskite fast ionic conductors materials (0.144 eV)[13]. These activation energies are much lower than we observe by total electrical impedance measurements for $Li_{1.5}La_{1.5}WO_6$ and $Li_{1.5}La_{1.5}TeO_6$. An individual muon samples a small, local region of a material and by implanting muons

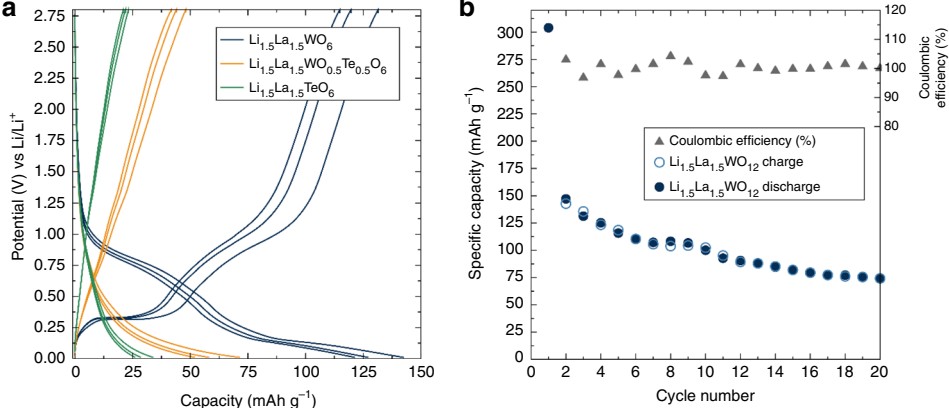

**Fig. 6 Battery cycling behaviour of $Li_{1.5}La_{1.5}MO_6$ Li-rich double perovskites. a** Galvanostatic cycling of $Li_{1.5}La_{1.5}MO_6$ double perovskites mixed with 5% carbon black and 5% PTFE binder, between 0.01 and 2.8 V vs Li at a specific current of 36 mA g$^{-1}$, where the first discharge has been omitted for clarity. **b** Charge/discharge capacity fading test for the $Li_{1.5}La_{1.5}WO_6$ material mixed with 5% carbon black and 5% PTFE binder at a current of 36 mA g$^{-1}$ between 0.01 and 2.8 V vs Li over 20 cycles. The initial high first discharge is due to the irreversible formation of the SEI layer caused by electrolyte decomposition.

throughout the material samples a volume-weighted average structure. As grain boundaries make up a minority of the volume of crystalline materials, such as these perovskites, the transport properties derived from muon measurements are strongly weighted towards the intra-grain transport properties. We have seen similar effects in fast Li$^+$ conducting garnet phases where the muon activation energy is much lower than that derived from impedance analysis of total conductivity[21].

$Li_{1.5}La_{1.5}WO_6$ shows ionic conductivity and low-voltage redox activity, suggesting possible application as an anode in Li-ion batteries. As CV measurements on $Li_{1.5}La_{1.5}TeO_6$ showed no redox activity, galvanostatic measurements on the Te analogue were carried out in order to serve as null measurements and evaluate the capacitive contribution from the carbon black additive used to increase the electronic conductivity of the electrode.

Galvanostatic testing of $Li_{1.5}La_{1.5}WO_6$ at a rate of 36 mA g$^{-1}$ gave the cycling profile and discharge capacity shown in Fig. 6. The first discharge capacity is irreversibly increased to above 300 mAh g$^{-1}$ due to electrolyte decomposition and formation of SEI. Subsequent cycles displayed reversible capacities of ~125 mAh g$^{-1}$ corresponding to an approximate 2e$^-$ transfer per formula unit of double perovskite material after carbon capacity subtraction determined from the $Li_{1.5}La_{1.5}TeO_6$ cycling capacity. In agreement with the CV results, a clear flat plateau is observed for $Li_{1.5}La_{1.5}WO_6$ at 0.35 V and the onset of a pseudo-plateau at ~0.7 V during charging. On the discharging cycles, a pseudo-plateau is observed below 0.9 V and a plateau at ~0.2 V. The mixed metal $Li_{1.5}La_{1.5}W_{0.5}Te_{0.5}O_6$ compound was found to have a discharge capacity intermediate between the parent $Li_{1.5}La_{1.5}MO_6$ compounds, confirming the redox activity experienced in these double perovskites is due to the W$^{6+}$ cations. Carbon coating of the $Li_{1.5}La_{1.5}WO_6$ particle surface was performed via a sucrose impregnation–carbonisation route in order to improve the performance and cyclability of the $Li_{1.5}La_{1.5}WO_6$ anode material. Carbon coating improves the electronic properties of the electrode composite and can also act as a buffer layer to protect from continuous side-reactions between the active materials surfaces and the electrolyte. The carbon-coating treatment resulted in an increased discharge capacity (Supplementary Fig. 9) with a value above 200 mAh g$^{-1}$ up to cycle 15, doubling that of the uncoated material's capacity of ≈100 mAh g$^{-1}$. Retention capacity is also greatly improved with the carbon-coating approach, with an increase from 53 to 85% on cycle 15 and from 41 to 62% at the end of cycle 20.

Assessing the redox activity of the tungsten analogue presented a challenge. XAS measurements of W L$_{III}$-edge (Supplementary Fig. 10) were performed on ex situ cycled samples but assigning oxidation state changes are difficult due to overlapping absorption edges. Only small differences in the relative intensity of the L$_{III}$ split peak were observable making it difficult to attribute these directly to oxidation changes of W. Difficulties in the analysis of W oxidation states by XAS has been previously reported in the literature[41]. Instead, we have used magnetometry to follow the reduction from diamagnetic W$^{6+}$ (5d$^0$) to paramagnetic species W$^{5+}$ or W$^{4+}$ during cycling. Measurements were conducted on two samples; a fully discharged sample and a sample charged back to 0.4 V (just above the first oxidation process). Magnetic measurements on the cycled materials point to oxidation state changes on the tungsten ions (Supplementary Fig. 11). For materials discharged to 0.01 V, Curie–Weiss paramagnetism indicative of fully localised unpaired electrons is observed arising from 1.43 μ$_B$ per formula unit, corresponding to the formation of 0.8 mol of W$^{5+}$ per formula unit. The observed Weiss constant of −120(2) K indicates strong antiferromagnetic coupling between neighbouring magnetic centres. In conjunction with the observed moment this shows that around 0.8 W$^{5+}$ cations per formula unit are interacting strongly, indicating short superexchange distance between these paramagnetic cations. This implies the W$^{5+}$ are evenly distributed throughout the perovskite phase that has a composition of $Li_{2.3}La_{1.5}WO_6$. From galvanostatic measurements, we expect to observe W$^{4+}$ at this voltage, suggesting that not all of the observed reduction leads to formation of localised, unpaired electrons. Instead, the excess 1.2 electrons per formula unit must lead to species that make no significant contribution to the Curie–Weiss behaviour. Delocalisation of some electrons leading to Pauli paramagnetism or a conversion reaction to a diamagnetic species would both be magnetically undetectable in the presence of the dominant signal from the W$^{5+}$. Cycling the material back to 0.4 V, beyond the first oxidation process, revealed a change in the magnetism of the cycled material with a large reduction of the paramagnetic moment to 0.84(3) μ$_B$ and a weak positive Curie constant of $\theta = +9(1)$ K. This indicates almost complete re-oxidation to W$^{6+}$.

Ex situ PXRD of fully discharged material revealed that the perovskite structure of $Li_{1.5}La_{1.5}WO_6$ is retained upon lithiation with a small displacement of the diffraction peaks towards higher d-spacing values, which is then reversed when the material is charged back to 2.8 V (Fig. 7). The absence of other phases in these diffraction patterns suggests intercalation as the main

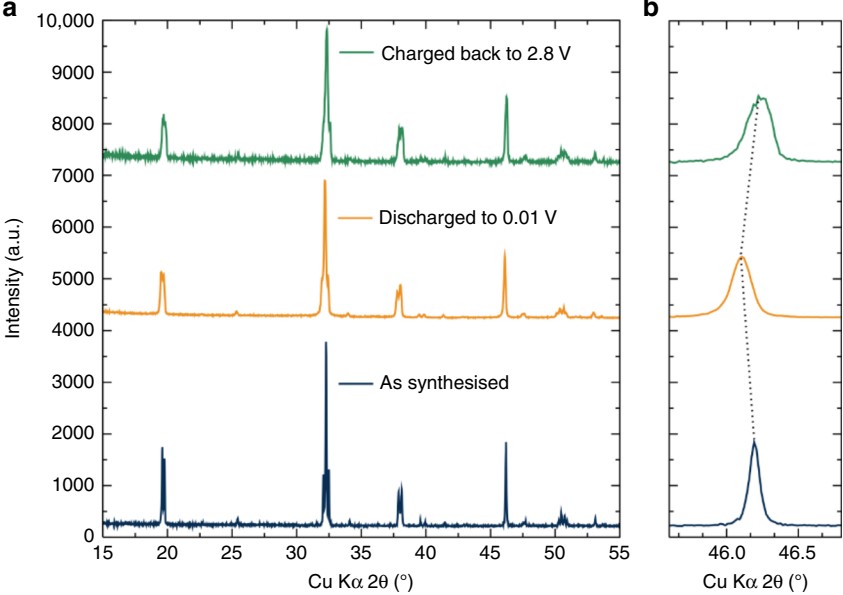

**Fig. 7 $Li_{1.5}La_{1.5}WO_6$ crystal lattice evolution upon cycling. a** Ex situ XRD patterns of the $Li_{1.5}La_{1.5}WO_6$ material as-synthesised, after fully electrochemical lithiation and after subsequent de-lithiation (20 cycles) (green line). **b** Magnification of the (220) peak evidences the reversible insertion and de-insertion of Li ions to and from the structure. The calculated volume expansion upon full lithiation is 0.2%.

mechanism for the observed electrochemical activity of this material. Interestingly, the small lattice parameter change indicates only a small volume increase from 245.016(3) Å before cycling to 245.50(2) Å post-lithiation. This 0.2% change is remarkably low compared to electrodes such as $LiFe_{0.8}Mn_{0.2}SO_4F$, which shows a volume change of 0.6% between the lithiated and delithiated phases[42,43], or $LiFePO_4$ where larger volume changes of 6.6% are noted[44]. The titanate anode material $Li_4Ti_5O_{12}$ also possesses a small volume expansion upon cycling (~0.4%), and intercalates three $Li^+$ per $Li_4Ti_5O_{12}$ unit giving a similar volumetric capacity to the $2e^-$ reversible cycling we observe per $Li_{1.5}La_{1.5}WO_6$[45]. This indicates that $Li_{1.5}La_{1.5}WO_6$ represents a promising anode material for Li-ion batteries, with very low (de) lithiation voltages and extremely low volume expansion upon cycling.

Analysis of diffraction data necessarily delivers a crystallographic model that is averaged over many unit cells and requires consideration of local atomic configurations. The disordered distribution of $La^{3+}$ and $Li^+$ identified in the Rietveld analysis, and the predicted sites for $Li^+$ intercalation were probed using computer simulation. Various cation-ordering schemes were computationally tested using a combination of site occupancy disorder[46] and molecular dynamics. These employed a supercell containing $Li_6La_6W_4O_{24}$ atoms with no symmetry constraints apart from periodic boundary conditions of *ca.* 7.9 Å in *x, y* and *z* directions. In $Li_{1.5}La_{1.5}WO_6$, $Li^+$ occupies a quarter of the *A* sites, and the simulations indicate that there is a small energy stabilisation (<10 kJ mol$^{-1}$ per $Li_{1.5}La_{1.5}WO_6$) for the lithium occupancy to be in adjacent *A* sites as shown in Fig. 8. With such a small difference in energy, we would not expect these weakly favoured local correlations in *A*-site occupancies to lead to a lowering of crystallographic symmetry from the observed monoclinic structure. These observations are in agreement with results for similar systems[19].

The effect of electrochemically inserting additional lithium was modelled by placing lithium in the largest voids in the structure and minimising the energy over the whole supercell (see Supplementary Note 1 for further details). The resultant structures show that reductive insertion of lithium is most readily achieved by

incorporation of additional lithium into the large *A* site interstices that already contain $Li^+$, with displacement of the $Li^+$ cations away from the centre of the hole to give lower coordination numbers and minimising the electrostatic repulsion arising from Li⋯Li interactions. The structure for $Li_{2.5}La_{1.5}WO_6$ contains three $Li^+$ cations occupying a single *A* site forming tetrahedral $LiO_4$ units and maintaining a minimum Li⋯Li separation of at least 2.44 Å. This stoichiometry corresponds to reduction to $W^{5+}$ and is in agreement with that indicated by the magnetometry measurements. Simulations can push the $Li^+$ content towards a limit of $Li_{3.0}La_{1.5}WO_6$ giving four $Li^+$ cations in the *A* site. This configuration involves somewhat shorter Li⋯Li distances, down to 2.29 Å, and considerable distortion of the $LiO_6$ and $WO_6$ units. Simulations on $Li_{1.5}La_{1.5}TeO_6$ show that, as expected, the $4d^{10}$ configuration of $Te^{6+}$ presents a barrier to reductive intercalation. Interestingly, Bader charge analysis of the oxidation states of tungsten in the $Li_{1.5}La_{1.5}WO_6$ systems shows a defined stepwise change from $Li_{1.50}La_{1.5}WO_6$ ($W^{6+}$, $d^0$) to $Li_{2.50}La_{1.5}WO_6$ ($W^{5+}$, $d^1$) to $Li_{3.00}La_{1.5}WO_6$ ($W^{4+}$ and $W^{5+}$ in 50:50 ratio, suggesting mixed $d^1$ and $d^2$ configurations) (Supplementary Fig. 12). Above $Li_{3.00}La_{1.5}WO_6$ the structure becomes heavily distorted and the electronic structure no longer follows a simple trend. Analogous calculations on the $Li_{1.5}La_{1.5}TeO_6$ structure (Supplementary Fig. 13 and Supplementary Table 7) have revealed large intercalation voltages for $Li^+$ intercalation together with reluctance of $Te^{6+}$ to form $Te^{5+}$, suggesting redox cycling of the Te analogue to be unlikely, as experimentally observed. Density of state analysis (Supplementary Fig. 14) also confirms the stability of this material against oxidation, with a large band gap of *ca.* 5 eV in the case of the $Li_{1.5}La_{1.5}TeO_6$ material, indicating high electrochemical stability as solid-state electrolyte.

In order to account for the observed two-electron transfer during cycling, we must consider alternative processes that may contribute to this increased capacity, such as conversion reaction to $Li_2O$[47]. From diffraction patterns of the cycled $Li_{1.5}La_{1.5}WO_6$ materials, there is no evidence of crystalline conversion products and scanning electron microscopy (SEM) images of post-cycled material do not suggest significant degradation of the electrode (Supplementary Fig. 15). To shed more light onto the possible

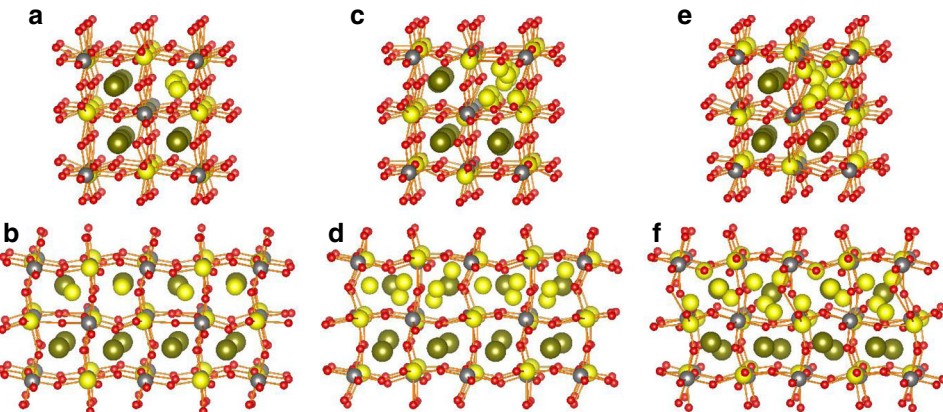

**Fig. 8 $Li_{1.5}La_{1.5}WO_6$ crystal structure evolution upon Li insertion. a, b** Orthogonal views of the simulated structure of the $Li_{1.5}La_{1.5}WO_6$ crystal structure (representing Li, W, La and O by yellow, grey, green and red spheres, respectively) illustrating preferential ordering of $Li^+$ cations into chains of face-sharing $A$ sites. **c, d** Insertion of additional lithium up to $Li_{2.5}La_{1.5}WO_6$ generates a cluster of three $LiO_4$ units within these $A$ sites, with a Li···Li cations separation distance of $\geq$2.44 Å. **e, f** Further reductive intercalation to $Li_{3.0}La_{1.5}WO_6$ leads to accommodation of four $Li^+$ cations in the $A$ site with shortened Li···Li distances and reduced screening as the oxide coordination around these $Li^+$ cations become increasingly distorted. A progressive distortion of the perovskite framework is observed with increasing insertion of $Li^+$ cations as it adjusts to accommodate the high concentration of $Li^+$ in a single $A$ site.

mechanism responsible for the additional capacity, we further examined the mixed metal $Li_{1.5}La_{1.5}W_{0.5}Te_{0.5}O_6$ phase. Interestingly, $Li_{1.5}La_{1.5}W_{0.5}Te_{0.5}O_6$ does not show a clear low redox couple in contrast to the well-defined peaks observed for $Li_{1.5}La_{1.5}WO_6$. Furthermore, this partial substitution of $W^{6+}$ by $Te^{6+}$ resulted in a reduction of the material specific capacity greater than that corresponding to decreasing concentration of redox active $W^{6+}$. Specifically, the observed capacity on the third cycle of *ca.* 120 mAh g$^{-1}$ in $Li_{1.5}La_{1.5}WO_6$ decreased to *ca.* 50 mAh g$^{-1}$ in $Li_{1.5}La_{1.5}W_{0.5}Te_{0.5}O_6$, and considering that the carbon black in the redox-inactive $Li_{1.5}La_{1.5}TeO_6$ provides near 25 mAh g$^{-1}$ (Fig. 6a), the reduction of the capacity when replacing half of $W^{6+}$ by $Te^{6+}$ in $Li_{1.5}La_{1.5}W_{0.5}Te_{0.5}O_6$ is almost fourfold (from ~95 to 25 mAh g$^{-1}$). In addition to this capacity loss, the long-range crystalline structure of $Li_{1.5}La_{1.5}W_{0.5}Te_{0.5}O_6$ is largely lost during the first battery discharge (Supplementary Fig. 16) suggesting a conversion-type process may be occurring for this material when reduced. The higher entropy of the resulting $Li_{1.5}La_{1.5}W_{0.5}Te_{0.5}O_6$ solid-solution phase could be the driving force underpinning the total macroscopic conversion, similarly reported for high entropy oxide materials in energy storage applications[48,49]. The CV data of the $Li_{1.5}La_{1.5}WO_6$ parent compound also show a lower reversibility of the higher voltage redox couple (around 1 V), an observation more commonly found in conversion processes[50–52]. These observations, combined with the results of magnetic measurements and structural simulations, suggest that the low-voltage redox processes observed for the pure tungsten material are a result of intercalation, while the behaviour at higher voltages may involve conversion processes, reminiscent of previous reports for other $W^{6+}$ oxides[53–56]. Similar combined mechanisms have been observed in $LiVO_3$, which displays both intercalation and conversion behaviour below 2 V, involving formation of $Li_{2.5}VO_3$ and conversion into metallic vanadium and $Li_2O$[57]. Other bimetallic vanadates have been reported to undergo an initial conversion process within the particles' surfaces followed by a intercalation process with subsequent $Li^+$ insertion[58–60]. Furthermore, unlike the $Li_{1.5}La_{1.5}TeO_6$ material, the $Li_{1.5}La_{1.5}WO_6$ material presents submicron-sized particles on the bulk surface, which could be more prone to conversion reactions (Supplementary Fig. 17).

A key motivation for employing solid electrolytes in all-solid-state batteries is their potential to safely facilitate metallic lithium

as an anode material[1,61]. To study the redox stability of $Li_{1.5}La_{1.5}TeO_6$ against lithium metal, an asymmetric cell was assembled using Li metal as a reference electrode and plasma deposited gold as a counter electrode. CV measurements at 80 °C reveal the ability of $Li_{1.5}La_{1.5}TeO_6$ to plate and strip lithium metal with no observable overpotential (Supplementary Fig. 18). Several peaks are observed in the reduction sweep of the first cycle, indicating the possible formation of an interfacial layer[62]. Subsequent cycles reveal that this interfacial layer bestows stability, with no further surface reactions noted. Interestingly, $Li_{1.5}La_{1.5}TeO_6$ displays good stability beyond 5 V making it a candidate for use in high-voltage cells. It should be noted that this is the first work on this novel material and that further optimisation, similar to that experienced by other solid-electrolyte systems (e.g., though microstructure optimisation for better Li wetting, protective layers to avoid reactions with electrodes)[63,64], is expected.

To optimise the ionic conductivity of the $Li_{1.5}La_{1.5}TeO_6$ material as a solid-state electrolyte, highly dense pellets were obtained by SPS. The relative density of the SPS pelleted material was greatly increased from *ca.* 76 to 98.1(1)%. Polarisation tests (Fig. 9a) indicate excellent compatibility and stability between Li metal and $Li_{1.5}La_{1.5}TeO_6$ during plating and stripping with Li electrodes. Impedance analysis of the $Li_{1.5}La_{1.5}TeO_6$ symmetric cells (Fig. 9b) reveals differences in the spectra observed for Li electrodes compared to Pt blocking electrodes. The latter contain a low frequency tail; this contrasts with the second semicircle observed when using Li electrodes, indicating the macroscopic mobility of $Li^+$. The small second semicircle observed when using Li electrodes is indicative of a low charge transfer resistance at the $Li/Li_{1.5}La_{1.5}TeO_6$ interface. Improvements in ionic conductivity are observed for the SPS treated $Li_{1.5}La_{1.5}TeO_6$ with a value of 0.12 mS cm$^{-1}$ at 124 °C, doubling that of the cold pressed material. The activation energy for $Li^+$ diffusion is also greatly reduced to 0.42(1) eV (Fig. 9b inset). This improvement in transport properties demonstrates the key role that pellet microstructure engineering has on the macroscopic conductivity measured by conventional electrochemical techniques, encouraging dedicated work to optimise this performance further. Subsequent improvement of conduction properties in novel oxide materials following their original report in the literature is often observed. For instance, benchmark Li-rich garnets oxides were originally reported to have ionic conductivities on the order of $10^{-6}$ S cm$^{-1}$ with

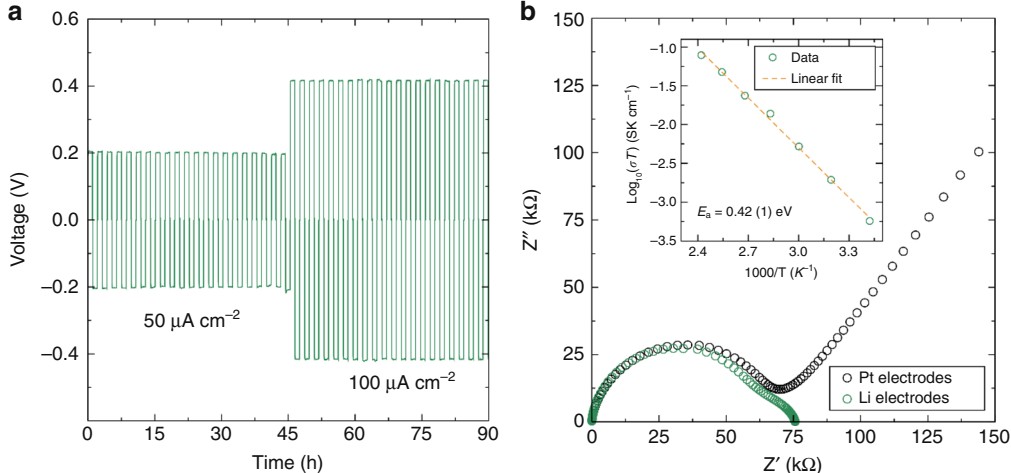

**Fig. 9 $Li_{1.5}La_{1.5}TeO_6$ stability against Li metal and transport properties. a** Polarisation test of a symmetric Li|$Li_{1.5}La_{1.5}TeO_6$|Li cell. The applied current densities were 50 and 100 µA cm$^{-2}$ at 80 °C. **b** EIS measurement at 19 °C for a Li|$Li_{1.5}La_{1.5}TeO_6$|Li and a Pt|$Li_{1.5}La_{1.5}TeO_6$|Pt symmetric cells. Inset shows the Arrhenius plot of conductivity measurements at different temperatures using Pt blocking electrodes.

activation energies in the 0.4–0.5 eV range, comparable with our novel $Li_{1.5}La_{1.5}TeO_6$ double perovskite, and improvements in the last decade have seen these values rise to above $10^{-3}$ S cm$^{-1}$ with activation energies below 0.2 eV[65–67]. The low local Li$^+$ activation energy below 0.2 eV and similar Li$^+$ diffusion coefficient obtained by µ$^+$SR here for the $Li_{1.5}La_{1.5}TeO_6$ material is comparable to that of the LLZO benchmark garnet electrolyte probed by the same technique, where again pellet microstructure greatly impacts the macroscopic transport properties[21]. This reinforces the scope for future improvements on the macroscopic transport properties of the $Li_{1.5}La_{1.5}TeO_6$ double perovskite reported here.

To evaluate the efficacy of the approach of crystal structure matching across the electrode–electrolyte interface, we tested the compatibility of the $Li_{1.5}La_{1.5}WO_6$ low-voltage negative electrode with the $Li_{1.5}La_{1.5}TeO_6$ solid-state electrolyte in a quasi-solid-state battery. This was carried out using a Li-metal half-cell comprising by the $Li_{1.5}La_{1.5}WO_6$ material as the electrode material and a hybrid electrolyte formulation $Li_{1.5}La_{1.5}TeO_6$:LiTFSI:Py$_{14}$TFSI (80:1:19$_{wt}$) [Py$_{14}$TFSI = 1-butyl-1-methylpyrrolidinium bis(trifluoromethylsulfonyl)imide] without the need for a separator or liquid electrolyte. The presence of the Py$_{14}$TFSI ionic liquid affords better wettability between the electrode and solid-electrolyte phases as well as lowering the resistance of Li$^+$ diffusion through the $Li_{1.5}La_{1.5}TeO_6$ solid-electrolyte under the conditions employed. CV analyses at 80 °C (Fig. 10) reveal the clear redox response of $Li_{1.5}La_{1.5}WO_6$, reminiscent of that observed for the conventional liquid electrolyte cell. The intense reduction peak at ~0.63 V is most likely due to the irreversible formation of SEI at the carbon black surface from the P$_{14}$TFSI ionic liquid, in agreement with that also observed for the conventional liquid electrolyte cell. The low-voltage redox peaks appear more defined and sharper when the $Li_{1.5}La_{1.5}TeO_6$ electrolyte is employed, indicating improved kinetics for Li$^+$ transference and insertion. The additional irreversibility observed during the first cycle of the $Li_{1.5}La_{1.5}TeO_6$ CV in the Li|$Li_{1.5}La_{1.5}TeO$|Au cell (Supplementary Fig. 18) could be arising from additional interphase formation or initial decomposition at the $Li_{1.5}La_{1.5}TeO_6$ interface, as observed in other solid-state electrolyte systems[68–70].

## Discussion

We have demonstrated that the tungsten and tellurium analogues of the Li-rich double perovskite family, $Li_{1.5}La_{1.5}MO_6$, are excellent candidate electrode and solid-electrolyte materials, respectively, for

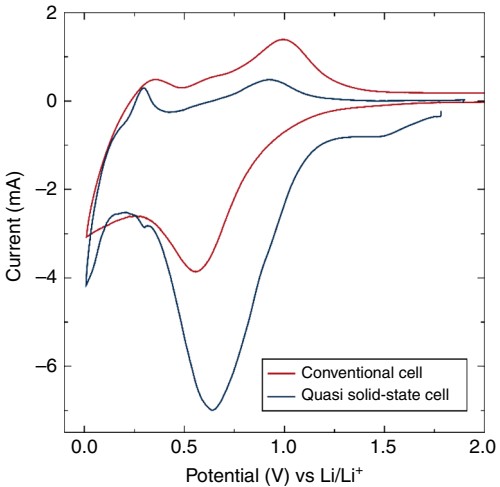

**Fig. 10 $Li_{1.5}La_{1.5}WO_6$–$Li_{1.5}La_{1.5}TeO_6$ compatibility in a single quasi all-solid-state battery.** CV of a Li half-cell formed by $Li_{1.5}La_{1.5}WO_6$:CB:PTFE (90:5:5$_{wt}$) as active material and a $Li_{1.5}La_{1.5}TeO_6$:LiTFSI:Py$_{14}$TFSI (80:1:19$_{wt}$) hybrid electrolyte tested against Li metal at 0.05 mV s$^{-1}$ in the voltage range 0.01–1.90 V at 80 °C. For comparison, a CV curve of the $Li_{1.5}La_{1.5}WO_6$ perovskite material in a conventional cell with 1 M LiPF$_6$ in EC: DMC (1:1$_{vol}$) liquid electrolyte cells has been included.

Li-ion batteries. The presence of Li ions in both *A* and *B* sites within the double perovskite framework enables Li-ion motion, and tailoring of the *B*-site cation directs the functionality towards a low-voltage negative electrode ($Li_{1.5}La_{1.5}WO_6$) or a solid-state electrolyte ($Li_{1.5}La_{1.5}TeO_6$). A detailed investigation into the redox mechanism of these materials unveils W$^{6+}$ as the redox active species during Li insertion, whilst Te$^{6+}$ confers a high redox stability, beyond 5 V, onto the perovskite crystal structure enabling its use as a safe alternative solid-state electrolyte. The promising diffusion properties of this new solid-electrolyte family presents an exciting opportunity for further performance optimisation through microstructure engineering and chemical composition exploration, reminiscent of previous oxide systems such as the Li-rich garnets or NASICON phases[6,34,71,72]. Interestingly, the combination of both Li-rich double perovskites into a single hybrid solid-state cell retains the electrode functionality and paves the way for tailored

isostructural design of solid-state battery components whereby interface compatibility can be finely tuned at the unit cell level.

## Methods

All reagent-grade chemicals employed for the synthesis of the Li-rich double perovskites were purchased from the following suppliers and used without further purification unless otherwise noted: LiOH·H$_2$O (98%) and La$_2$O$_3$ (99%) from Sigma Aldrich, WO$_3$ (99.95%) and TeO$_2$ (99.978%) were purchased from Alfa Aesar.

For the microwave-assisted synthesis of the Li$_{1.5}$La$_{1.5}$WO$_6$, Li$_{1.5}$La$_{1.5}$TeO$_6$ and Li$_{1.5}$La$_{1.5}$W$_{0.5}$Te$_{0.5}$O$_6$ double perovskites, stoichiometric amounts of La$_2$O$_3$ (previously dried at 900 °C for 24 h), WO$_3$ and/or TeO$_2$, and a 10% excess of LiOH·H$_2$O ($^7$LiOH·H$_2$O for NPD studies) were ball milled for 30 min at a vibrational frequency of 20 Hz in a stainless-steel jar. Subsequently, the fine powder was pelleted under uniaxial load of 3 t. The pelleted material was heated at 700 °C for 6 h in a 2.45 GHz CEM Phoenix hybrid microwave furnace for the decomposition of the precursor materials. Subsequently, the material was reground and pelleted for a second heat treatment carried out in air at 900 °C for 6 h in the same microwave furnace. The last treatment consisted in 1 h at 1000 °C of the repelleted material in the same hybrid microwave furnace. In every calcination, the heating rate was held at 2 °C min$^{-1}$ to reduce lithium evaporation.

Powder XRD was employed for the assessment of the purity and study of the crystal parameters for compounds prepared. A PANalytical X'Pert PRO Diffractometer was used for this purpose using Cu-Kα radiation in the 2θ range 15–130° with a nominal scan rate of 800 s per step and a step size of 0.016° at room temperature.

NPD patterns used for Rietveld refinements were collected in Bank 3 (2θ = 35.26°) and Bank 5 (2θ = 91.34°) at the GEM instrument for the Li$_{1.5}$La$_{1.5}$WO$_6$ material, and Bank 3 (2θ = 51.99°) and Bank 5 (2θ = 146.94°) at the Polaris instrument for the Li$_{1.5}$La$_{1.5}$TeO$_6$ analogue at the ISIS pulsed neutron and muon source at the Rutherford Appleton Laboratory, UK. The data were collected over a time-of-flight region 0.25–23 ms (Polaris) and 0.9–21.5 ms (GEM). The sample, ca. 1–3 g, was placed in an 11-mm-diameter cylindrical vanadium can and loaded into the beamline. The data were collected at room temperature. The broad incident pulsed neutron flux was narrowed in a 100 K methane moderator giving a peak flux at λ = 2 Å, prior to sample scattering[73,74]. Rietveld refinements against XRD and NPD data were performed with the Generalized Structure Analysis System (GSAS)[75], along with the graphical user interface EXPGUI[76], by means of a least square approach.

In order to analyse the size and morphology of the synthesised particles, SEM images were acquired with a Carl Zeiss Sigma microscope. All samples were ground, and a tiny amount of the fine powder was deposited over a carbon-taped sample holder. Subsequently the sample was Au-coated and ready for analysis.

EDX spectra were recorded using an Oxford Instruments Energy 250 energy-dispersive spectrometer system. Copper tape was employed as a standard for calibration and the voltage of the incident beam was 25 keV.

Induced coupled plasma-mass spectroscopy (ICP-MS) analyses were performed on an Agilent 7700 ICP-MS instrument. Approximately 6 mg of sample was dissolved in 50 mL of 2% HNO$_3$ solution in deionized water and immersed into an ultrasonic bath for 5 min prior to the measurements.

EIS AC measurements with Pt electrode were performed on a Solartron 1260 Impedance Analyzer in the frequency range of 1–10$^6$ Hz and a temperature range between RT and 400 °C in 50 °C steps using as-synthesised pelleted materials. In order to enhance the connection between the pellet and the electrodes, a suspension of 0.5–5 μm platinum particles in n-butyl acetate was prepared and a few drops of this suspension were deposited on both surfaces of the pelletized material. The electrodes consisted of square pieces of 0.025-mm-thickness platinum foil, connected through 0.127-mm-diameter platinum wire to the device.

CV and galvanostatic cycling measurements were conducted in a BioLogic VSP3 potentiostat using two electrode Swagelok type cells. The electrode material was formed by a mixture of the double perovskite material (90%), electronically conductive carbon black Ketjenblack EC600JD (AkzoNobel) (5%) and PTFE (polytetrafluoroethylene) (Fischer Scientific) (5%) as binder to the electrode mixture. A Whatman glass microfibre filter (GF/D grade) was used as a separator with 1 M LiPF$_6$ in ethylene carbonate and dimethyl carbonate 1:1 v/v (Solvionic) as the electrolyte and a 10-mm-diameter circular piece of Li metal of 0.75 mm thickness (Sigma Aldrich) as the reference and counter electrode. C-coated Li$_{1.5}$La$_{1.5}$WO$_6$ (targeting 10%$_{wt}$ and resulting in a 6%$_{wt}$ real content as calculated from EA) was produced by sucrose route previously employed in our group[77]. In brief, Li$_{1.5}$La$_{1.5}$WO$_6$ as-synthesised material was mixed with sucrose in a 50:50 (%$_{vol}$) ethanol:water solution. The resulting suspension was sonicated for 30 min and subsequently heated until the solvent evaporated. The mixture was then dried under vacuum at 80 °C for 12 h before carbonisation in a tube furnace under flowing Ar gas for 3 h at 700 °C. For the Li$_{1.5}$La$_{1.5}$TeO$_6$ perovskite material, an asymmetrical cell composed of pelletised Li$_{1.5}$La$_{1.5}$TeO$_6$ material sandwiched between Li metal as reference and counter electrode and sputtered gold as working electrode was mounted on a Swagelok cell at 80 °C for CV analyses on the same VSP3 instrument at a voltage scanning speed of 0.05 mV s$^{-1}$. In the case of the polarisation test and EIS of the symmetrical Li$_{1.5}$La$_{1.5}$TeO$_6$ cell with Li-metal electrodes, the same set-up was employed by replacing the Au layer with a Li-metal electrode.

SPS experiments were performed in an FCT HP D 25 SPS furnace. Powder samples were loaded into a cylindrical graphite die with a 16-mm-inner diameter, lined with thin graphite foil. Rapid heating was controlled by a thermocouple inserted within the die for the duration of the experiment. The set-up was held under constant uniaxial pressure of 50 MPa whilst DC current pulses were used to heat the sample at a constant rate of 50 °C/min via Joule heating between particles. The target temperature was 1090 °C with a dwell time of 5 min once this was reached. After the experiment, the sample was allowed to cool naturally. The material was then removed from the die, the surface graphite removed, and polished using sandpaper of up to 2500 grit to obtain smooth pellets of ~12 mm diameter and 1.5 mm thickness. AC impedance and galvanostatic cycling measurements of SPS samples were performed on a Biologic VSP potentiostat in the frequency range of 1 Hz to 7 MHz with a 50 mV voltage perturbation. Gas deposition was employed to sputter coat electrodes in Pt using a Polaron SC7640 with a sputtering time of 200 s, a current of 20 mA and an Ar pressure of ~0.02 mbar. The same pellet was used for measurements with lithium electrodes, whereby the pellet was polished to remove the Pt coating and transferred to an Ar-filled glovebox. The pellet was sandwiched between two lithium foils (Sigma Aldrich, 0.38 mm) and assembled within a Swagelok cell. The surface of each lithium foil was scraped using a stainless-steel blade to ensure optimal contact with the pellet.

Density values were obtained through helium gas displacement pycnometry using a Micromeritics AccuPyc II 1340 system.

Raman data were acquired in a Horiba Jobin Yvon LabRAM HR system equipped with a Ventus 532 laser. Spectra were acquired in the 50–4000 cm$^{-1}$ Raman shift range with a 532-nm-wavelength laser with a 100 mW power. The light beam was masked to the appropriated level in order to obtain data intensity in a measurable range.

The XAS data were collected in the B18 beamline at Diamond Source of Light synchrotron. For the data acquisition, a few milligrams (between 10 and 100 mg) of the as-synthesised materials were mixed with cellulose fibre (ca. 100 mg) and compacted into a 10-mm-diameter thin pellet and stored in an aluminium plastic bag pocket under Ar inert atmosphere. Samples were mounted into a holder and exposed to the synchrotron X-ray radiation emitted by a bending magnet source which is monochromatised and focused by a vertically collimating Si mirror, a water-cooled Si(111) and Si(311) double crystal monochromator and a focusing double toroidal mirror. The data were collected in the transmission mode using three ionisation chambers mounted in series for simultaneous measurements on the sample and a tungsten foil as reference. Scans of ca. 3–5 m were collected over the desired energy range and merging of three consecutive scans was performed to obtain precise data sets. The data were processed and normalised using the Athena software package using edge step normalisation. For fits of the EXAFS data, the Artemis software package was employed. The Artemis software allows one to perform FEFF calculations of the theoretical EXAFS from the crystal data obtained by diffraction methods to obtain the individual scattering pathways, which then can be fit to the experimental EXAFS in order to compared atomic distances in the local and the average structures. Through the incorporated IFEFFIT algorithm, the data were fitted by a least-squares procedure until the best possible fit was achieved[78].

μ$^+$SR studies were performed using the EMU instrument at the ISIS pulsed muon facility[79]. The powdered sample, ca. 1.5 g, was packed into a disk of 30 mm diameter and 1.5 mm thickness and sealed in a titanium sample holder where the front window was made of 25-μm-thickness titanium foil. 3.2 MeV spin-polarised positive muons were implanted into the sample and the outcoming positrons were detected by 96 scintillator segments grouped in two circular arrays. The temperature was controlled from 100 K up to a maximum of 600 K by a hot stage attached to a closed cycle refrigerator and the measurements were acquired at three different applied longitudinal magnetic fields (0, 5 and 10 G). A 20 G transverse magnetic field was also applied for the initial asymmetry calibration. The data fits were carried out through the WiMDA data analysis programme[80].

Magnetic measurements were conducted by using a Quantum Design MPMS-XL SQUID magnetometer to measure ca. 20 mg of cycled material sample contained in a gelatine capsule. Data were collected in an applied field after cooling in either the measuring field or in zero applied field and were not corrected for diamagnetism. Susceptibility data were fitted using the Curie–Weiss law in the temperature range 150–290 K.

DFT calculations employed the Vienna Ab initio Simulation Package with the projector augmented wave method[81] and the PBEsol functional[82]. An additional Dudarev U$_{eff}$ parameter, obtained from the materials project database, was used to afford a more accurate representation of any d-orbital electrons of tungsten[83] (GGA+U calculations: https://materialsproject.org/wiki/index.php/GGA%2BU_calculations). A 700 eV plane-wave energy cutoff was used and spin polarisation was introduced. A varying Γ-centred Monkhorst–Pack k-point mesh was employed depending on the size of the simulation cell. The Li$_6$La$_6$W$_4$O$_{24}$ system used a converged k-point mesh of 2 × 2 × 2. Lattice vectors were allowed to vary during minimisations and a force tolerance of 0.01 eV Å was applied for convergence. The minimisations are followed by a single point energy calculation with the HSE06 functional at the Γ-point. The use of the hybrid HSE06 functional better accounts for localisation of d electrons and therefore was used for Bader charge and

electronic structure analysis. Further details on computational analyses are found in the Supplementary Information.

## Data availability

The data sets generated and analysed during the current study are available from the corresponding author on reasonable request. The data sets generated using the ISIS neutron and muon source are available from ISIS Neutron and Muon Source Data Catalogue using https://doi.org/10.5286/ISIS.E.RB1620255 and https://doi.org/10.5286/ISIS.E.RB1720318 for muon and neutron data sets, respectively.

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

## Acknowledgements

The authors gratefully acknowledge technical support from Michael Beglan and Peter Chung at the University of Glasgow. The authors also thank the EPSRC for grant funding (SUPERGEN Challenge grant on "Design and high throughput microwave synthesis of Li-ion battery materials", EP/N001982/1); the support of the Faraday Institution (SOL-BAT, Grant No. FIRG007); the STFC for beamtime allocation through the GEM, Polaris and EMU beamlines at the ISIS neutron and muon source and the Diamond Light Source for XAS beamtime through the Energy Materials Block Allocation Group organised by Alan Chadwick and Giannantonio Cibin (proposal 25120); computational facilities such as the ARCHER supercomputer through membership of the UK's HPC Materials Chemistry Consortium (funded by EPSRC Grants EP/L000202/1 and EP/R029431/1), the "Hydra" High Performance System at Loughborough University and the use of Athena as part of the HPC Midlands+ consortium (EPSRC, EP/P020232/1); the Universities of Sheffield, Glasgow, Strathclyde and Loughborough for support and the use of facilities; and the School of Chemistry at Glasgow for PhD studentship funding.

## Competing interests
The authors declare no competing interests.
