## [Peer Review File · Nature Communications]

Reviewers' comments:

Reviewer #1 (Remarks to the Author):

This manuscript describes the Li-rich perovskite materials with W and Te for battery applications. Two different chemistries provided in this manuscript, but the manuscript is not well constructed and confusing. I am not supportive for the publication to Nature Communications. Specific comments are described in the following;

- (1) Figure 6 reversible capacity must be shown. In the caption "anomalously high first discharge" is used. This is because of electrolyte decomposition and not anomalous capacity.
- (2) Figure 8 and Li insertion into the W phase. Description is not clear. Structural model after Li insertion would be added, instead of (c) and (e), in which only local environment is shown.
- (3) Small volume change for the W phase on lithiation is described. However, cyclability is not acceptable for battery applications, indicating that electrolyte decomposition (and SEI formation) is more important for the electrode materials with low electrochemical potential.
- (4) Line 396-405, LiVO_3 shows phase transition into a rocksalt phase after Li insertion, forming Li_2VO_3 . But, this is not conversion reaction, and the sample shows cation migration on discharge.
- (5) Figure 9, too large polarization is observed even at 80 °C, and this phase is difficult to use for all solid-state Li-ion batteries.
- (6) Small reversible capacity is observed for the Te phase, but the amorphous phase is formed after reduction. What is the driving force for the phase transition?
- (7) Figure 10, explanation is not clear. W and Te, which one is used? Very confusing. Moreover, no progress is found for HSE, and a much larger irreversible capacity is observed for HSE.

Reviewer #2 (Remarks to the Author):

The article describes a comprehensive characterisation of a new family of perovskite materials that can be designed to act as solid lithium ion conductors or active materials. This strategy is illustrated with the full characterisation of two model compounds, containing W and Te respectively. A careful quantitative analysis of the materials properties has been achieved with a range of techniques, including XRD, Raman, ICP-MS, EXAFS, electrochemical characterisation and muon spin relaxation measurements. As the authors briefly mention, the evaluation of the conductivity from impedance measurements in Li-electrolyte-Li cells could be underestimated because of the effect of the charge transfer resistance that is developed at the Li-electrolyte interphases. Therefore, the evaluation of conductivity from impedance measurements with blocking electrodes appears more reliable, and indeed, the authors report higher conductivity values obtained by this method.

Reviewer #4 (Remarks to the Author):

In this work, the authors conducted a comprehensive study for the Li-rich double perovskite family $\text{Li}_{1.5}\text{La}_{1.5}\text{MO}_6$ ($M = \text{W}^{6+}, \text{Te}^{6+}$). $\text{Li}_{1.5}\text{La}_{1.5}\text{WO}_6$ is a low-voltage anode candidate, and $\text{Li}_{1.5}\text{La}_{1.5}\text{TeO}_6$ can be used as electrolyte with high redox stability up to 5V. The structure characterizations, performance measurements, and the hypothetical mechanism of cycling demonstrate that the double perovskite family compounds are of interest to the energy material development.

While potentially interesting, I cannot recommend the publication of this article in Nature Communications at this juncture. Here are the major concerns:

1. Neither the anode material nor solid electrolyte versions of the material are particularly high performing. Due to the use of W, the specific capacity of the anode is low. The conductivity of the Te

material is not high enough to be a good electrolyte (0.058 mS/cm). While it is possible that the Te material can be further improved with processing, this was not demonstrated.

2. In order to explain the two-electron extraction per unit $\text{Li}_{1.5}\text{La}_{1.5}\text{WO}_6$, the authors proposed possible conversion process during cycling. This also explains the poor reversibility of the cycling, which again detracts from the use of this material as an anode.

3. The author performed some DFT simulations of the W material, but none for the Te material. Given that the Te material is somewhat more interesting, this absence of DFT data on the Te material is puzzling. There is no analysis of reactions of the Te material with Li or at high voltages, even though the claim is that the Te material is stable up to 5V. Even a simple band gap calculation would be sufficient to establish whether this is even theoretically possible. But it would be ideal if a proper study of the reaction with Li be done. There are many works in the literature demonstrating such analysis.

Response to reviewers' comments

Reviewers' comments:

Reviewer #1 (Remarks to the Author):

Comment 1: Figure 6 reversible capacity must be shown. In the caption “anomalously high first discharge” is used. This is because of electrolyte decomposition and not anomalous capacity.

Response 1: Figure 6 has been updated to include charge and discharge specific capacities, as well as coulombic efficiency for each charge-discharge cycle. We have also updated the caption to indicate the high initial discharge capacity is due to SEI formation caused by electrolyte decomposition.

Figure 6: (a) Galvanostatic cycling of Li_{1.5}La_{1.5}MO₆ double perovskites mixed with 5% carbon black and 5% PTFE binder, between 0.01 V and 2.8 V vs Li at a specific current of 36 mA g⁻¹, where the first discharge has been omitted for clarity. (b) Charge/discharge capacity fading test for the Li_{1.5}La_{1.5}WO₆ material mixed with 5% carbon black and 5% PTFE binder at a current of 36 mA g⁻¹ between 0.01 V and 2.8 V vs Li over 20 cycles. The initial high first discharge is due to the irreversible formation of the SEI layer caused by electrolyte decomposition.

Comment 2: Figure 8 and Li insertion into the W phase. Description is not clear. Structural model after Li insertion would be added, instated of (c) and (e), in which only local environment is shown.

Response 2: We thank the reviewer for their suggestion to clarify Figure 8. We have updated it accordingly and include now larger cell views for the Li-intercalated phases with an updated description.

Figure 8: Orthogonal views a) and b) of the simulated structure of the $\text{Li}_{1.5}\text{La}_{1.5}\text{WO}_6$ crystal structure (representing Li, W, La and O by yellow, grey, green and red spheres respectively) illustrating preferential ordering of Li^+ cations into chains of face-sharing A sites. c) and d) Insertion of additional lithium up to $\text{Li}_{2.5}\text{La}_{1.5}\text{WO}_6$ generates a cluster of three LiO_4 units within these A sites, with a $\text{Li}\cdots\text{Li}$ cations separation distance of ≥ 2.44 Å. e) and f) Further reductive intercalation to $\text{Li}_{3.0}\text{La}_{1.5}\text{WO}_6$ leads to accommodation of four Li^+ cations in the A site with shortened $\text{Li}\cdots\text{Li}$ distances and reduced screening as the oxide coordination around these Li^+ cations become increasingly distorted. A progressive distortion of the perovskite framework is observed with increasing insertion of Li^+ cations as it adjusts to accommodate the high concentration of Li^+ in a single A site.

Comment 3: *Small volume change for the W phase on lithiation is described. However, cyclability is not acceptable for battery applications, indicating that electrolyte decomposition (and SEI formation) is more important for the electrode materials with low electrochemical potential.*

Response 3: We acknowledge the concerns of the reviewer regarding cyclability and agree that electrolyte decomposition plays a key role in the performance of negative electrodes at low voltages windows below the electrochemical stability of the electrolyte. Several strategies have been explored in the literature to increase the performance and reversibility of low voltage negative electrodes in Li-ion batteries, including; cycling at an specific initial current rate for an optimised SEI formation[1], the use of stabilising additives in the electrolyte formulation[2] or active material surface functionalisation or coating,[3] among others. While the main aim of our work is to demonstrate how judicious tailoring of the chemistry and crystal structure of Li-rich double perovskites can provide new materials wherein the unusual Li^+ arrangement opens up its functionality for reversible energy storage, we have optimised the battery performance and cyclability of our $\text{Li}_{1.5}\text{La}_{1.5}\text{WO}_6$ material by particle surface carbon coating. This carbon coating process has been carried out via a sucrose impregnation-calcination route previously employed in our group,[4] which allows for an improved electronic conductivity of the electrode mixture and can also act as protective layer to the $\text{Li}_{1.5}\text{La}_{1.5}\text{WO}_6$ particles towards continuous reaction with the electrolyte. This approach results in increased discharge capacity [Figure S9] with a value above 200 mAh g^{-1} for the carbon-coated up to \approx cycle 15, doubling that of the uncoated material with a value near 100 mAh g^{-1} . Retention capacity is also greatly improved with the carbon coating approach, with an increase from 53 to 85% on cycle 15 and from 41 to 62% at the end of cycle 20.

The following text and figure have been included in the main text and supporting information, respectively:

Carbon coating of the $\text{Li}_{1.5}\text{La}_{1.5}\text{WO}_6$ particle surface was performed via a sucrose impregnation-carbonisation route in order to improve the performance and cyclability of the $\text{Li}_{1.5}\text{La}_{1.5}\text{WO}_6$ anode material. Carbon coating improves the electronic properties of the electrode composite and can also act as a buffer layer to protect from continuous side-reactions between the active materials surfaces and the electrolyte. The carbon-coating treatment resulted in an increased discharge capacity [Figure S9] with a value above 200 mAh g^{-1} for the carbon-coated up to cycle 15, doubling that of the uncoated material with a value near 100 mAh g^{-1} . Retention capacity is also greatly improved with the carbon-coating approach, with an increase from 53 to 85% on cycle 15 and from 41 to 62% at the end of cycle 20.

C-coated $\text{Li}_{1.5}\text{La}_{1.5}\text{TeO}_6$ (targeting 10%_{wt} and resulting in a 6%_{wt} real content as calculated from EA) was produced by sucrose route previously employed in our group.⁷⁴ In brief, LLWO as-synthesised material was mixed with sucrose in a 50:50 (%_{vol}) ethanol:water solution. The resulting suspension was sonicated for 30 min and subsequently heated until the solvent evaporated. The mixture was then dried under vacuum at 80 °C for 12 hours before carbonization in a tube furnace under flowing Ar gas for 3 h at 700 °C.

Figure S9: Discharge capacities observed for $\text{Li}_{1.5}\text{La}_{1.5}\text{WO}_6$ Li-rich double perovskite anode material with (blue circles) and without carbon coating (orange triangles) cycled at 17 mA g^{-1} .

With the present improvement introduced in this revision, we have demonstrated a large enhancement on the reversibility and discharge capacity of the $\text{Li}_{1.5}\text{La}_{1.5}\text{WO}_6$ material as negative electrode. As the reviewer points out, the main reason for the detriment of the cyclability is centred at the liquid electrolyte decomposition at low voltages. In this regard, we are providing here a novel class of materials with both redox-active electrode material and solid-state electrolyte with compatibility between them, contributing to the pursuit of an all-solid-state Li-ion battery which eventually would allow overcoming the safety and cyclability issues arising from the liquid electrolytes.

[1] *ACS Appl. Mater. Interfaces*, **2019**, 11, 34796; [2] *Energy Environ. Sci.*, **2016**, 9, 1955; [3] *J. Mater. Chem. A*, **2020**, 8, 3606; [4] *Chem. Commun.*, **2016**, 52, 9028

Comment 4: Line 396-405, LiVO_3 shows phase transition into a rocksalt phase after Li insertion, forming Li_2VO_3 . But, this is not conversion reaction, and the sample shows cation migration on discharge.

Response 4: We regret that the wording of this paragraph could have created confusion. In the work entitled “Unusual Conversion-type Lithiation in LiVO_3 Electrode for Lithium-Ion Batteries”, the authors describe how “a two-phase (nucleation/growth type) conversion reaction is followed along with a structural disintegration; the $\text{Li}_{2.5}\text{VO}_3$ phase decomposes into metallic vanadium and Li_2O .” The confusion here could arise from the assignment of the first reduction reaction observed between our $\text{Li}_{1.5}\text{La}_{1.5}\text{WO}_6$ material and Li^+ attributed to conversion processes occurring on the particle surface, with the subsequent reduction process attributed to Li^+ intercalation into the particle bulk. Typical conversion reactions in W^{6+} oxides occur in the 0.6 – 1.5 V vs Li/Li^+ voltage window, [5-8] matching well with the 0.7/1.0 V redox couple observed in our $\text{Li}_{1.5}\text{La}_{1.5}\text{WO}_6$ Li-rich double perovskite. Other bimetallic vanadates have been also reported to follow a similar mechanism during reduction, involving initial surface conversion which is then followed by an intercalation process, schematically represented in Figure R1 below.[9-11] Additionally, SEM images have revealed the presence of submicron-sized particulates at the $\text{Li}_{1.5}\text{La}_{1.5}\text{WO}_6$ material, which could further corroborate this attribution of a surface predisposition to conversion reactions. To clarify this point, the following text and figure have been included in the main manuscript and supporting information, respectively:

These observations, combined with the results of magnetic measurements and structural simulations, suggest that the low voltage redox processes observed for the pure tungsten material are a result of intercalation, while the behaviour at higher voltages may involve conversion processes, reminiscent of previous reports for other W^{6+} oxides.⁵³⁻⁵⁶ Similar combined mechanisms have been observed in LiVO_3 which displays both intercalation and conversion behaviour below 2 V, involving formation of $\text{Li}_{2.5}\text{VO}_3$ and conversion into metallic vanadium and Li_2O .⁵⁷ Other bimetallic vanadates have been reported to undergo an initial conversion process within the particles’ surfaces followed by a intercalation process with subsequent Li^+ insertion.⁵⁸⁻⁶⁰ Furthermore, unlike the $\text{Li}_{1.5}\text{La}_{1.5}\text{TeO}_6$ material, the $\text{Li}_{1.5}\text{La}_{1.5}\text{WO}_6$ material presents submicron-sized particles on the bulk surface which could be more prone to conversion reactions [Figure S17].

Figure S17: SEM images of as-synthesised $\text{Li}_{1.5}\text{La}_{1.5}\text{TeO}_6$ (a,c) and $\text{Li}_{1.5}\text{La}_{1.5}\text{WO}_6$ (b, d) at different magnifications. Large bulk materials decorated with submicron-sized particles are observed for $\text{Li}_{1.5}\text{La}_{1.5}\text{WO}_6$ perovskite. Similar bimodal particle distributions are not observed for the $\text{Li}_{1.5}\text{La}_{1.5}\text{TeO}_6$ material.

Figure R1: Schematic representation of electrochemical reaction of bimetallic vanadates with Li^+ where conversion and intercalation mechanism coexist. Figure reproduced from [5].

[5] *ACS Appl. Mater. Interfaces*, **2015**, 7, 7635; [6] *Electrochim. Acta*, **2016**, 187, 329; [7] *Appl. Surf. Sci.*, **2010**, 256, 2447; [8] *Electrochim. Acta*, **2015**, 163, 132; [9] *Adv. Energy Mater.* **2019**, 9, 1803324; [10] *ACS Nano* **2014**, 8, 5, 4474; [11] *Small* **2017**, 13, 1603140

Comment 5: Figure 9, too large polarization is observed even at 80 oC, and this phase is difficult to use for all solid-state Li-ion batteries.

Response 5: Following this comment from the reviewer, we have optimised the microstructure of the pelletised solid-electrolyte material through spark plasma sintering (SPS). We have achieved notable improvements in the ionic conductivity properties of the $Li_{1.5}La_{1.5}TeO_6$ material which is also accompanied by a dramatic decrease in symmetric cell polarisation. Specifically, the ionic conductivity is doubled, reaching a value of 1.2 mS cm^{-1} at $124 \text{ }^\circ\text{C}$ with cell polarisation decreased to 0.2 V at $50 \text{ } \mu\text{A cm}^{-2}$ (corresponding to a polarisation of $4 \text{ mV } \mu\text{A}^{-1}$, on the SPS treated sample) compared to the 4.2 V at $10 \text{ } \mu\text{A cm}^{-2}$ (corresponding to a polarisation of $420 \text{ mV } \mu\text{A}^{-1}$), i.e., a decrease of **100-fold in the polarisation**. The activation energy required for Li^+ diffusion is also decreased to a value of $0.42(1) \text{ eV}$ when using the SPS technique, down from $0.68(2) \text{ eV}$ for the untreated material.

The following text has been included in the main manuscript and Figure 9 have been updated as follows:

To optimise the ionic conductivity of the $Li_{1.5}La_{1.5}TeO_6$ material as a solid-state electrolyte, highly dense pellets were obtained by spark plasma sintering (SPS). The relative density of the SPS pelletised material was greatly increased from ca. 76% to 98.1(1)%. Polarisability tests [Figure 9(a)] indicate excellent compatibility and stability between Li metal and $Li_{1.5}La_{1.5}TeO_6$ during plating and stripping with Li electrodes. Impedance analysis of the $Li_{1.5}La_{1.5}TeO_6$ symmetric cells [Figure 9(b)] reveals differences in the spectra observed for Li electrodes compared to Pt blocking electrodes. The latter contain a low frequency tail; this contrasts with the second semicircle observed when using Li electrodes, indicating the macroscopic mobility of Li^+ . The small second semicircle observed when using Li electrodes is indicative of a low charge transfer resistance at the Li/LLTeO interface. Improvements in ionic conductivity are observed for the SPS treated $Li_{1.5}La_{1.5}TeO_6$ with a value of 0.12 mS

cm^{-1} at 124 °C, double that of the cold pressed material. The activation energy for Li^+ diffusion is also greatly reduced to 0.42(1) eV [Figure 9(b) inset].

Figure 9: (a) Polarisation test of a symmetric $\text{Li}|\text{Li}_{1.5}\text{La}_{1.5}\text{TeO}_6|\text{Li}$ cell. The applied current densities were $50 \mu\text{A cm}^{-2}$ and $100 \mu\text{A cm}^{-2}$ at 80 °C. (b) EIS measurement at 19 °C for a $\text{Li}|\text{Li}_{1.5}\text{La}_{1.5}\text{TeO}_6|\text{Li}$ and a $\text{Pt}|\text{Li}_{1.5}\text{La}_{1.5}\text{TeO}_6|\text{Pt}$ symmetric cells. Inset shows the Arrhenius plot of conductivity measurements at different temperatures using Pt blocking electrodes.

Through the application of SPS, we have overcome a major obstacle hindering the full potential of solid-state electrolytes, namely grain boundary resistances. Microengineering of solid-electrolytes is a common route to improve their macroscopic conduction properties, as for example reported in the related $\text{Li}_{3x}\text{La}_{0.67-x}\text{TiO}_3$ perovskite to enhance its ionic conductivity.[12,13] There are additional barriers that could impede the realisation of the full potential on our novel $\text{Li}_{1.5}\text{La}_{1.5}\text{TeO}_6$ material such as the presence of single-atom-layer traps consisting of defect loops that prevent large volumes of materials from participating in ionic transport, as recently reported for the $\text{Li}_{3x}\text{La}_{0.67-x}\text{TiO}_3$ perovskite system.[14] We are working on doping strategies which could have the potential to disrupt possible SALT formations and further increase the ionic conductivity properties.[15] Preliminary work has shown [Figure R2], however, that this is not a trivial exercise, highlighting the complexity and uniqueness of the Li-rich double perovskite structure and compositions presented here.

Figure R2: PXRD patterns of $\text{Li}_{1.5}\text{La}_{1.5}\text{TeO}_6$ and Ca-substituted $\text{Li}_2\text{LaCa}_{0.5}\text{TeO}_6$ material where the introduction of Ca^{2+} dopant results in the presence of secondary phases.

[12] *J. Mater. Chem. A*, **2017**, 5, 6257; [13] *Mater. Horiz.*, **2019**, 6, 871; [14] *Nat. Commun.*, **2020**, 11, 1828; [15] *J. Membr. Sci.*, **2019**, 582, 194

Comment 6: *Small reversible capacity is observed for the Te phase, but the amorphous phase is formed after reduction. What is the driving force for the phase transition?*

Response 6: We thank the reviewer for their comment. We understand from this comment that the reviewer here is referring to the Te/ W mixed phase, $\text{Li}_{1.5}\text{La}_{1.5}\text{W}_{0.5}\text{Te}_{0.5}\text{O}_6$, where both W and Te are present in the crystal structure of the same phase forming a solid-solution, where we performed *ex-situ* PXRD showing amorphisation after reduction. This solid solution was synthesised primarily to demonstrate that the redox activity in the $\text{Li}_{1.5}\text{La}_{1.5}\text{WO}_6$ material originates from the W cations, which is evidenced by a reduction in capacity with decreasing W content. As previously observed for bimetallic vanadates,[16-18] we hypothesise that the surface of the material is prone to conversion reactions owing to the presence of submicron-sized particles on the material surface as observed by SEM [Figure S17]. Surface reactivity has also been reported for other oxide materials, such as Ni-rich NMC layered oxides where surface reconstruction and phase transformation distinct from the bulk material has been observed.[19-21] The observation of amorphisation in the $\text{Li}_{1.5}\text{La}_{1.5}\text{W}_{0.5}\text{Te}_{0.5}\text{O}_6$ material further corroborates the propensity of these materials to undergo surface conversion processes. While the exact driving force for the full amorphisation of the $\text{Li}_{1.5}\text{La}_{1.5}\text{W}_{0.5}\text{Te}_{0.5}\text{O}_6$ remains to be fully determined and is the focus of ongoing work, we hypothesise that the lattice energy of the solid-solution could be higher than the parent compounds due to Te and W disorder on the B-sites making this phase more susceptible to conversion reactions. Similar behaviour has been observed in high entropy oxides, where the original crystal structure is conserved (at the nanoscale) while serving as

a permanent host matrix for conversion cycles when reduced by Li^+ insertion.[22,23] The transition from a *B*-site ordered material to a *B*-site disordered crystal structure and the increase in phase entropy could be driving force for the full macroscopic conversion of the $\text{Li}_{1.5}\text{La}_{1.5}\text{W}_{0.5}\text{Te}_{0.5}\text{O}_6$ material.

The following sentence has been added to the main manuscript, following this reviewers interesting point:

The higher entropy of the resulting $\text{Li}_{1.5}\text{La}_{1.5}\text{W}_{0.5}\text{Te}_{0.5}\text{O}_6$ solid-solution phase could be the driving force underpinning the total macroscopic conversion, similarly reported for high entropy oxide materials in energy storage applications.^{48,49}

[16] *Adv. Energy Mater.* **2019**, *9*, 1803324; [17] *ACS Nano* **2014**, *8*, 5, 4474; [18] *Small* **2017**, *13*, 1603140; [19] *Nat. Energy*, **2016**, *1*, 15004; [20] *Nat. Commun.*, **2014**, *5*, 3529; [21] *ACS Appl. Energy Mater.*, **2020**, *3*, 4799; [22] *Nat. Commun.*, **2018**, *9*, 3400; [23] *ACS Appl. Mater. Interfaces*, **2020**, *12*, 23860

Comment 7: *Figure 10, explanation is not clear. W and Te, which one is used? Very confusing. Moreover, no progress is found for HSE, and a much larger irreversible capacity is observed for HSE.*

Response 7: We thank the reviewer for raising this clarity issue in Figure 10, describing the use of our $\text{Li}_{1.5}\text{La}_{1.5}\text{TeO}_6\text{:LiTFSI:Py}_{14}\text{TFSI}$ hybrid electrolyte. The concept behind this test is the use of $\text{Li}_{1.5}\text{La}_{1.5}\text{WO}_6$ as an electrode material and $\text{Li}_{1.5}\text{La}_{1.5}\text{TeO}_6$ as the solid-state electrolyte material in the same battery cell, with the ultimate goal of demonstrating an all-solid-state battery. Here, we form a pseudo all-solid-state battery in which we add a small amount of ionic liquid to the $\text{Li}_{1.5}\text{La}_{1.5}\text{TeO}_6$ solid-state electrolyte phase to facilitate interfacial Li^+ transport. The reviewer is correct in pointing out the larger degree of irreversibility in the initial cycling of the cell. This is likely due to the additional contribution from interphase formation or initial decomposition at the electrolyte interface, as previously reported for other solid-state electrolytes,[27-29] and also noted in the first cycle CV of the $\text{Li}_{1.5}\text{La}_{1.5}\text{TeO}_6$ material (Figure S18). However, this proof-of-concept experiment demonstrates the interface compatibility between the $\text{Li}_{1.5}\text{La}_{1.5}\text{WO}_6$ electrode and $\text{Li}_{1.5}\text{La}_{1.5}\text{TeO}_6$ electrolyte. To simplify the interpretation of this test, we have now removed the carbon black blank cell dataset. While the main objective here was to demonstrate compatibility between materials with similar crystal chemistries in the same battery cell, it should be noted that quasi-solid-state cells using hybrid electrolytes also have reported advantages and applications.[24-26]

[24] *J. Power Sources*, **2019**, *438*, 226985; [25] *J. Membr. Sci.*, **2020**, *603*, 117820; [26] *J. Electrochem. Soc.*, **2020**, *167*, 040511; [27] *Joule*, **2018**, *2*, 1991; [28] *Adv. Energy Mater.*, **2019**, *9*, 1901810; [29] *Nat. Rev. Mater.*, **2020**, *5*, 105

To clarify this point following this reviewer suggestion, we have revised the text describing the quasi all-solid-state battery analyses:

To evaluate the efficacy of the approach of crystal structure matching across the electrode-electrolyte interface, we tested the compatibility of the $\text{Li}_{1.5}\text{La}_{1.5}\text{WO}_6$ low voltage negative electrode with the $\text{Li}_{1.5}\text{La}_{1.5}\text{TeO}_6$ solid-state electrolyte in a quasi-solid-state battery. This was carried out using a Li-metal half-cell comprising by the $\text{Li}_{1.5}\text{La}_{1.5}\text{WO}_6$ material as the electrode material and a hybrid electrolyte formulation $\text{Li}_{1.5}\text{La}_{1.5}\text{TeO}_6\text{:LiTFSI:Py}_{14}\text{TFSI}$ (80:1:19%_{wt}) [$\text{Py}_{14}\text{TFSI}$ = 1-butyl-1-methylpyrrolidinium bis(trifluoromethylsulfonyl)imide] without the need for a separator or liquid electrolyte. The presence of the $\text{Py}_{14}\text{TFSI}$ ionic

liquid affords better wettability between the electrode and solid-electrolyte phases as well as lowering the resistance of Li^+ diffusion through the $\text{Li}_{1.5}\text{La}_{1.5}\text{TeO}_6$ solid-electrolyte under the conditions employed. CV analyses at 80 °C [Figure 10] reveal the clear redox response of $\text{Li}_{1.5}\text{La}_{1.5}\text{WO}_6$, reminiscent of that observed for the conventional liquid electrolyte cell. The intense reduction peak at ~ 0.63 V is most likely due to the irreversible formation of SEI at the carbon black surface from the P_{14}TFSI ionic liquid, in agreement with that also observed for the conventional liquid electrolyte cell. The low voltage redox peaks appear more defined and sharper when the $\text{Li}_{1.5}\text{La}_{1.5}\text{TeO}_6$ electrolyte is employed, indicating improved kinetics for Li^+ transference and insertion. The additional irreversibility observed during the first cycle of the $\text{Li}_{1.5}\text{La}_{1.5}\text{TeO}_6$ CV in the $\text{Li}|\text{Li}_{1.5}\text{La}_{1.5}\text{TeO}_6|\text{Au}$ cell (Figure S18) could be arising from additional interphase formation or initial decomposition at the $\text{Li}_{1.5}\text{La}_{1.5}\text{TeO}_6$ interface, as observed in other solid-state electrolyte systems.⁶⁷⁻⁶⁹

Figure 10: CV of a Li half-cell formed by $\text{Li}_{1.5}\text{La}_{1.5}\text{WO}_6$:CB:PTFE (90:5:5 %wt) as active material and a $\text{Li}_{1.5}\text{La}_{1.5}\text{TeO}_6$:LiTFSI:Py₁₄TFSI (80:1:19 %wt) hybrid electrolyte (HE) tested against Li metal at 0.05 mV s^{-1} in the voltage range 0.01 V - 1.90 V at 80 °C. For comparison, a CV curve of the $\text{Li}_{1.5}\text{La}_{1.5}\text{WO}_6$ perovskite material in a conventional cell with 1 M LiPF_6 in EC:DMC (1:1 %vol) liquid electrolyte (LE) cells has been included.

Reviewer #2 (Remarks to the Author):

The article describes a comprehensive characterisation of a new family of perovskite materials that can be designed to act as solid lithium ion conductors or active materials. This strategy is illustrated with the full characterisation of two model compounds, containing W and Te respectively. A careful quantitative analysis of the materials properties has been achieved with a range of techniques, including XRD, Raman, ICP-MS, EXAFS, electrochemical characterisation and muon spin relaxation measurements. As the authors briefly mention, the evaluation of the conductivity from impedance measurements in Li-electrolyte-Li cells could be underestimated because of the effect of the charge transfer resistance that is developed at the Li-electrolyte interphases. Therefore, the evaluation of conductivity from impedance measurements with blocking electrodes appears more reliable, and indeed, the authors report higher conductivity values obtained by this method.

Response: We thank the reviewer for their appreciation to our manuscript and positive response towards its publication.

Reviewer #4 (Remarks to the Author):

Comment 1: *Neither the anode material nor solid electrolyte versions of the material are particularly high performing. Due to the use of W, the specific capacity of the anode is low. The conductivity of the Te material is not high enough to be a good electrolyte (0.058 mS/cm). While it is possible that the Te material can be further improved with processing, this was not demonstrated.*

Response 1: To address these concerns, we have explored several routes to improve the performance of these novel materials. In the case of the $\text{Li}_{1.5}\text{La}_{1.5}\text{WO}_6$ anode material, carbon coating via a sucrose impregnation-calcination methodology has been carried out in order to improve the electronic conduction properties of the electrode composite as well as to provide a protective layer to the $\text{Li}_{1.5}\text{La}_{1.5}\text{WO}_6$ particles towards any detrimental side reactions with the electrolyte. This carbon-coating approach resulted in an increased discharge capacity [Figure S9] with a value above 200 mAh g^{-1} for the carbon-coated sample up to cycle 15, doubling that of the uncoated material with a value near 100 mAh g^{-1} . Retention capacity is also greatly improved with the carbon coating approach, with an increase from 53 to 85% on cycle 15 and from 41 to 62% at the end of cycle 20.

The following text and figure have been included in the main text and supporting information, respectively:

Carbon coating of the $\text{Li}_{1.5}\text{La}_{1.5}\text{WO}_6$ particle surface was performed via a sucrose impregnation-carbonisation route in order to improve the performance and cyclability of the $\text{Li}_{1.5}\text{La}_{1.5}\text{WO}_6$ anode material. Carbon coating improves the electronic properties of the electrode composite and can also act as a buffer layer to protect from continuous side-reactions between the active materials surfaces and the electrolyte. The carbon-coating treatment resulted in an increased discharge capacity [Figure S9] with a value above 200 mAh g^{-1} for the carbon-coated up to cycle 15, doubling that of the uncoated material with a value near 100 mAh g^{-1} . Retention capacity is also greatly improved with the carbon-coating approach, with an increase from 53 to 85% on cycle 15 and from 41 to 62% at the end of cycle 20.

C-coated $\text{Li}_{1.5}\text{La}_{1.5}\text{TeO}_6$ (targeting 10%_{wt} and resulting in a 6%_{wt} real content as calculated from EA) was produced by sucrose route previously employed in our group.⁷⁴ In brief, LLWO as-synthesised material was mixed with sucrose in a 50:50 (%_{vol}) ethanol:water solution. The resulting suspension was sonicated for 30 min and subsequently heated until the solvent evaporated. The mixture was then dried under vacuum at 80 °C for 12 hours before carbonization in a tube furnace under flowing Ar gas for 3 h at 700 °C.

Figure S9: Discharge capacities observed for $\text{Li}_{1.5}\text{La}_{1.5}\text{WO}_6$ Li-rich double perovskite anode material with (blue circles) and without carbon coating (orange triangles) cycled at 17 mA g^{-1} .

To investigate further improvements to the theoretical capacity, we have followed a cation substitution strategy for our $\text{Li}_{1.5}\text{La}_{1.5}\text{WO}_6$ perovskite by substituting La^{3+} cations with lighter Ca^{2+} cations [Figure R3a]. The perovskite framework, however, rearranges to a conventional double perovskite structure without Li^+ occupying vacant A-sites, resulting in a considerable capacity decrease to below 50 mAh g^{-1} [Figure R3b]. This highlights the unique structure of the $\text{Li}_{1.5}\text{La}_{1.5}\text{WO}_6$ composition and Li^+ arrangement as enabler of the functional properties as Li-ion battery material.

Figure R3: a) PXRD patterns of $\text{Li}_{1.5}\text{La}_{1.5}\text{WO}_6$ and Ca-substituted LiCaLaWO_6 where the double perovskite structure is retained. b) Galvanostatic cycling of the LiCaLaWO_6 double perovskite at 38 mA g^{-1} between 0.01 and 2.8 V. After the large discharge capacity on the first cycle attributed to SEI formation, the capacity decreases to a final value near 30 mAh g^{-1} .

With regards to the novel $\text{Li}_{1.5}\text{La}_{1.5}\text{TeO}_6$ solid-state electrolyte, we have optimised the microstructure of the pelletised material through spark plasma sintering (SPS). We have observed considerable improvements in the ionic conductivity properties of the $\text{Li}_{1.5}\text{La}_{1.5}\text{TeO}_6$ material which is also accompanied by a dramatic decrease on the symmetric cell polarisation. The ionic conductivity is doubled, reaching a value of 1.2 mS cm^{-1} at $124 \text{ }^\circ\text{C}$, with cell polarisation decreased to 0.2 V at $50 \text{ } \mu\text{A cm}^{-2}$ (corresponding to a polarisation of $4 \text{ mV } \mu\text{A}^{-1}$) on the SPS treated sample compared to the 4.2 V at $10 \text{ } \mu\text{A cm}^{-2}$ (corresponding to a polarisation of $420 \text{ mV } \mu\text{A}^{-1}$) on the untreated sample, i.e., a **100-fold decrease in polarisation**. The activation energy required for Li^+ diffusion is also decreased to $0.42(1) \text{ eV}$ when using the SPS technique, down from $0.68(2) \text{ eV}$ for the untreated material.

The following text have been included in the main text and Figure 9 have been updated as follows:

To optimise the ionic conductivity of the $\text{Li}_{1.5}\text{La}_{1.5}\text{TeO}_6$ material as a solid-state electrolyte, highly dense pellets were obtained by spark plasma sintering (SPS). The relative density of the SPS pelletised material was greatly increased from ca. 76% to 98.1(1)%. Polarisation tests [Figure 9(a)] indicate excellent compatibility and stability between Li metal and $\text{Li}_{1.5}\text{La}_{1.5}\text{TeO}_6$ during plating and stripping with Li electrodes. Impedance analysis of the $\text{Li}_{1.5}\text{La}_{1.5}\text{TeO}_6$ symmetric cells [Figure 9(b)] reveals differences in the spectra observed for Li electrodes compared to Pt blocking electrodes. The latter contain a low frequency tail; this contrasts with the second semicircle observed when using Li electrodes, indicating the macroscopic mobility of Li^+ . The small second semicircle observed when using Li electrodes is indicative of a low charge transfer resistance at the Li/LLTeO interface. Improvements in ionic conductivity are observed for the SPS treated $\text{Li}_{1.5}\text{La}_{1.5}\text{TeO}_6$ with a value of 0.12 mS cm^{-1} at $124 \text{ }^\circ\text{C}$, double that of the cold pressed material. The activation energy for Li^+ diffusion is also greatly reduced to $0.42(1) \text{ eV}$ [Figure 9(b) inset].

Figure 9: (a) Polarisation test of a symmetric $\text{Li}|\text{Li}_{1.5}\text{La}_{1.5}\text{TeO}_6|\text{Li}$ cell. The applied current densities were $50 \text{ } \mu\text{A cm}^{-2}$ and $100 \text{ } \mu\text{A cm}^{-2}$ at $80 \text{ }^\circ\text{C}$. (b) EIS measurement at $19 \text{ }^\circ\text{C}$ for a $\text{Li}|\text{Li}_{1.5}\text{La}_{1.5}\text{TeO}_6|\text{Li}$ and a $\text{Pt}|\text{Li}_{1.5}\text{La}_{1.5}\text{TeO}_6|\text{Pt}$ symmetric cells. Inset shows the Arrhenius plot of conductivity measurements at different temperatures using Pt blocking electrodes.

A doping strategy could be a plausible route to further increase the intrinsic conductivity of the $\text{Li}_{1.5}\text{La}_{1.5}\text{TeO}_6$ material. This has been demonstrated in a wide range of solid electrolytes, including the aliovalent substitutions of La^{3+} with M^{2+} cations, such as Ca^{2+} , to increment the

concentration of mobile Li^+ . [30,31] Attempts to dope with Ca^{2+} our $\text{Li}_{1.5}\text{La}_{1.5}\text{TeO}_6$ have demonstrated again the peculiarity and uniqueness of our Li-rich double perovskite composition, where the formation of secondary phases is observed [Figure R2]. Further experimental and computational analyses are on-going to find suitable cations to dope both $\text{Li}_{1.5}\text{La}_{1.5}\text{MO}_6$ and $\text{Li}_{1.5}\text{La}_{1.5}\text{WO}_6$ materials, while retaining the distinctive Li^+ arrangement that enables their functionality as battery materials.

Figure R2: PXRD patterns of $\text{Li}_{1.5}\text{La}_{1.5}\text{TeO}_6$ and Ca-substituted $\text{Li}_2\text{LaCa}_{0.5}\text{TeO}_6$ materials, where the introduction of Ca^{2+} dopant results in the presence of secondary phases.

[30] *APL Materials*, **2018**, 6, 060702; [31] *Dalton Trans.*, **2017**, 46, 9415

Comment 2: In order to explain the two-electron extraction per unit $\text{Li}_{1.5}\text{La}_{1.5}\text{WO}_6$, the authors proposed possible conversion process during cycling. This also explains the poor reversibility of the cycling, which again detracts from the use of this material as an anode.

Response 2: Following this reviewer's point, we have performed carbon-coating of the $\text{Li}_{1.5}\text{La}_{1.5}\text{WO}_6$ material to provide a protective layer that could improve conversion processes occurring at the particle's surface, as already introduced in our previous response to this reviewer and in response to comment 3 of Reviewer #1, where an improved capacity and reversibility were achieved.

SEM images of post-cycled $\text{Li}_{1.5}\text{La}_{1.5}\text{WO}_6$ [Figure S15] demonstrate the robustness of these particles, with no noticeable pulverisation that is typically noted for materials that undergo full conversion with high irreversibility. [32] We also note that while conversion electrodes typically suffer from cyclability issues due to large volume changes and pulverisation, improvements have been made in recent years to alleviate these following different strategies, such as surface coatings. [33] Our work therefore provides an unique avenue for the development of a family of materials where judicious choice of the metal cations in the crystal framework can tailor the electronic properties that governs functionality. In this case, we demonstrate both a redox active electrode material and a solid-state electrolyte that can advance in the pursuit of an all all-solid-state Li-ion battery.

[32] *Adv. Energy Mater.*, **2017**, 7, 1700715; [33] *Energy Environ. Sci.*, **2017**, 10, 435

Comment 3: The author performed some DFT simulations of the W material, but none for the Te material. Given that the Te material is somewhat more interesting, this absence of DFT data on the Te material is puzzling. There is no analysis of reactions of the Te material with Li or at high voltages, even though the claim is that the Te material is stable up to 5V. Even a simple band gap calculation would be sufficient to establish whether this is even theoretically possible. But it would be ideal if a proper study of the reaction with Li be done. There are many works in the literature demonstrating such analysis.

Response 3: We thank the reviewer for this suggestion. We have performed DFT simulations on the $\text{Li}_{1.5}\text{La}_{1.5}\text{TeO}_6$ material to complement our experimental observations [Figures S13 and S14]. A band gap near 5 eV was calculated [Figure S14], which is on a par with Density Of States (DOS) calculations for high-performance garnet solid electrolyte materials[34,35] and considerable superior to other benchmark solid-state electrolytes such as sulphides, NASICON or even the related $\text{Li}_{3x}\text{La}_{0.67-x}\text{TiO}_3$ perovskite where the presence of redox active Ti cations results in reductive reaction with Li metal. Most important is the direct observation of electrochemical stability of our material in direct contact with Li metal and cycling beyond 5 V with absence of anodic peaks up to 5 V, indicating the absence of oxidative decomposition of the $\text{Li}_{1.5}\text{La}_{1.5}\text{TeO}_6$ electrolyte [Figure S18].

Figure S14: Electronic DOS for a) $\text{Li}_{1.5}\text{La}_{1.5}\text{WO}_6$ and b) $\text{Li}_{1.5}\text{La}_{1.5}\text{TeO}_6$. A clear lack of significant available states for redox reactions of the $\text{Li}_{1.5}\text{La}_{1.5}\text{TeO}_6$ materials is observed, in contrast of those observed below 3 eV for the $\text{Li}_{1.5}\text{La}_{1.5}\text{WO}_6$ analogue, indicating a large band gap and wide electrochemical stability window.

Regarding the stability at low voltages, analogous calculations to those presented for the $\text{Li}_{1.5}\text{La}_{1.5}\text{WO}_6$ material have now been performed and included in the manuscript [Figure S13 and Table S6], following this reviewer's suggestion. In contrast with the redox mechanism for Li^+ intercalation on the $\text{Li}_{1.5}\text{La}_{1.5}\text{WO}_6$ material, large intercalation voltages are found for the $\text{Li}_{1.5}\text{La}_{1.5}\text{TeO}_6$ together with a reluctance of Te^{6+} to form Te^{5+} , suggesting redox cycling of the Te analogue to be unlikely, as experimentally observed. The W analogue, however, is capable of accessing the intermediate W^{5+} oxidation state, allowing a uniform step change in oxidation states ($\text{W}^{4+} \rightarrow \text{W}^{5+} \rightarrow \text{W}^{6+}$) throughout the material during lithium deintercalation, again, as experimentally observed.

Figure S13: Bader charge calculated for computationally lithiated LLTeO structures. LLTeO prefers a reduction from 6+ to 4+ before a second Te species begins reduction.

To study the redox behaviour in the Te analogues, the tungsten ions from the optimised LLWO analogues were replaced by tellurium ions and reminimised using the same procedure. The intercalation voltages and nominal redox couples for LLTeO are listed in Table S7.

Table S7: The structural formula, intercalation voltage and nominal redox couple of lithiated LLTeO.

Structural formula	Number of Li ions in cell	Intercalation voltage (V)	Nominal Redox couple
Li _{1.50} La _{1.5} TeO ₆	6		
Li _{1.75} La _{1.5} TeO ₆	7	0.59	Te ⁶⁺ - Te ⁵⁺
Li _{2.00} La _{1.5} TeO ₆	8	1.64	Te ⁵⁺ - Te ⁴⁺
Li _{2.25} La _{1.5} TeO ₆	9	0.68	Te ⁶⁺ - Te ⁵⁺
Li _{2.50} La _{1.5} TeO ₆	10	1.91	Te ⁵⁺ - Te ⁴⁺
Li _{2.75} La _{1.5} TeO ₆	11	1.08	Te ⁶⁺ - Te ⁵⁺
Li _{3.00} La _{1.5} TeO ₆	12	1.53	Te ⁵⁺ - Te ⁴⁺

The following text has been included in the main manuscript:

Analogous calculations on the Li_{1.5}La_{1.5}TeO₆ structure [Figure S13, Table S7] have revealed large intercalation voltages for Li⁺ intercalation together with reluctance of Te⁶⁺ to form Te⁵⁺, suggesting redox cycling of the Te analogue to be unlikely, as experimentally observed. Density of state calculations [Figure S14] have also confirmed the stability of this material against oxidation, with a large band gap of ca. 5 V in the case of the Li_{1.5}La_{1.5}TeO₆ material, indicating high electrochemical stability as solid-state electrolyte.

Reviewer #1 (Remarks to the Author):

Well updated.

Reviewer #2 (Remarks to the Author):

I thank the authors for attempting to address the comments. Despite the great effort and improvement in this manuscript, the point remains that this material is not particularly high performing. Even for the SPS material, the conductivity is 0.12 mS/cm at 124C, which means at room-temperature the conductivity will be too low of interest. This is also reflected in the high activation energy of 0.42 eV.

This is clearly a good quality work. But insufficient in novelty or performance for Nature Communications.

REVIEWERS' COMMENTS

Reviewer #1 (Remarks to the Author):

Well updated.

Reviewer #2 (Remarks to the Author):

I thank the authors for attempting to address the comments. Despite the great effort and improvement in this manuscript, the point remains that this material is not particularly high performing. Even for the SPS material, the conductivity is 0.12 mS/cm at 124C, which means at room-temperature the conductivity will be too low of interest. This is also reflected in the high activation energy of 0.42 eV.

This is clearly a good quality work. But insufficient in novelty or performance for Nature Communications.

Response to Reviewer#2

We thank the reviewers for their appreciation of our additional work and the quality of our manuscript.

Here, we respond to the comments made by Reviewer 2, whose recommendation has been made based on the macroscopic descriptor for the transport properties and their suggestion of insufficient novelty.

In this work, we are reporting a new family of Li-rich double perovskites with battery applications not previously reported in the literature. We have undertaken a comprehensive approach to characterising the physical properties that enable their electrochemical functionality. The $\text{Li}_{1.5}\text{La}_{1.5}\text{TeO}_6$ Li-rich double perovskite, reported for the first time here, is the first demonstrated example of a Li-rich double perovskite with Li-ion conduction, stability and compatibility with a Li metal anode, and an electrochemical stability window *surpassing* 5 V vs Li/Li⁺.

We would like to point out the variability of macroscopic conductivity as a descriptor of performance, which does not always reveal intrinsic transport properties of materials. For example, in our previous work on the known LLZO benchmark garnet material [*J. Mater. Chem. A*, **2016**, 4, 1729-1736] we demonstrated the promising microscopic properties of the material, that are being matched years later in recent studies [*ACS Appl. Mater. Interfaces*, **2020**, 12, 29, 32806–32816], and the impact of the pellet microstructure when employing conventional electrochemical techniques. Our $\text{Li}_{1.5}\text{La}_{1.5}\text{TeO}_6$ Li-rich double perovskite presented here possesses comparable local transport properties with activation energy *below* 0.2 eV and Li⁺ diffusion in the order of $10^{-11} \text{ cm}^2 \text{ s}^{-1}$, reinforcing the scope for further optimisation at the macroscopic level.

To address the current macroscopic conductivity performance and scope for optimisation compared to other benchmark materials, such as garnets, we have added the following discussion in the main manuscript:

“This improvement in transport properties demonstrates the key role that pellet microstructure engineering has on the macroscopic conductivity measured by conventional

electrochemical techniques, encouraging dedicated work to optimise this performance further. Subsequent improvement of conduction properties in novel oxide materials following their original report in the literature is often observed. For instance, benchmark Li-rich garnets oxides were originally reported to have ionic conductivities on the order of 10^{-6} S cm⁻¹ with activation energies in the 0.4 - 0.5 eV range, comparable with our novel $\text{Li}_{1.5}\text{La}_{1.5}\text{TeO}_6$ double perovskite, and improvements in the last decade have seen these values rise to above 10^{-3} S cm⁻¹ with activation energies below 0.2 eV.⁶⁵⁻⁶⁷ The low local Li^+ activation energy below 0.2 eV and similar Li^+ diffusion coefficient obtained by $\mu^+\text{SR}$ here for the $\text{Li}_{1.5}\text{La}_{1.5}\text{TeO}_6$ material is comparable to that of the LLZO benchmark garnet electrolyte probed by the same technique, where again pellet microstructure greatly impacts the macroscopic transport properties.²¹ This reinforces the scope for future improvements on the macroscopic transport properties of the $\text{Li}_{1.5}\text{La}_{1.5}\text{TeO}_6$ double perovskite reported here.”